# Symbolic Graphics Programming with Large Language Models

## Abstract

Large language models (LLMs) excel at program synthesis, yet their ability to produce symbolic graphics programs (SGPs) that render into precise visual content remains underexplored. We study symbolic graphics programming, where the goal is to generate an SGP from a natural-language description. This task also serves as a lens into how LLMs understand the visual world by prompting them to generate images rendered from SGPs. Among various SGPs, our paper sticks to scalable vector graphics (SVGs), as they are widely used and can be easily rendered into images. We begin by examining the extent to which LLMs can generate SGPs. To this end, we introduce SGP-GenBench, a comprehensive benchmark covering object fidelity, scene fidelity, and compositionality (attribute binding, spatial relations, numeracy). On SGP-GenBench, we discover that frontier proprietary models substantially outperform open-source models, and performance correlates well with general coding capabilities. Motivated by this gap, we are interested in how to improve LLMs' ability to generate SGPs. We propose a reinforcement learning (RL) with verifiable rewards approach, where a format-validity gate ensures renderable SVG, and a cross-modal reward aligns text and the rendered image via strong vision encoders (*e.g.*, SigLIP for text-image and DINO for image-image). Applied to Qwen-2.5-7B, our method substantially improves SVG generation quality and semantics, achieving performance on par with frontier systems. We further analyze training dynamics, showing that RL induces (i) finer decomposition of objects into controllable primitives and (ii) contextual details that improve scene coherence. Our results demonstrate that symbolic graphics programming offers a precise and interpretable lens on cross-modal grounding, while reinforcement learning with cross-modal rewards provides a scalable way for injecting visual knowledge into LLMs.

## 1 Introduction

Accurately describing a complex scene using natural language is inherently difficult. Natural language often leaves room for ambiguity, lacking the precise spatial, geometric, and structural details needed to fully capture a visual scene. While such flexibility is advantageous for everyday communication, it poses significant challenges for tasks that demand unambiguous, executable specifications of visual content. Symbolic graphics programs (SGPs) offer a promising alternative, as they encode scenes as structured, formal representations that can be deterministically rendered into graphics content like images or 3D objects. By bridging the gap between abstract linguistic descriptions and concrete visual representations, SGPs provide a means to represent scenes with both precision and compositionality.

Motivated by advances in program synthesis with large language models (LLMs) (Austin et al., 2021; Nijkamp et al., 2022), we study their ability to perform symbolic graphics programming, which generates SGPs given a natural language description. Given that LLMs are pretrained on large corpora of code, we expect them to be capable of understanding SGPs as a specialized class of programs. Qiu et al. (2025) and Zou et al. (2024a) have shown that LLMs possess semantic understanding of SGPs. Building on these findings, we extend the research question to whether LLMs can generate SGPs. Unlike question answering over SGPs in Qiu et al. (2025), SGP generation demands a more precise understanding of the correspondence between semantics and programs. Moreover, this task serves as a lens into how well LLMs can both understand and synthesize the visual world by producing graphics objects (*e.g.*, images) rendered from SGPs.

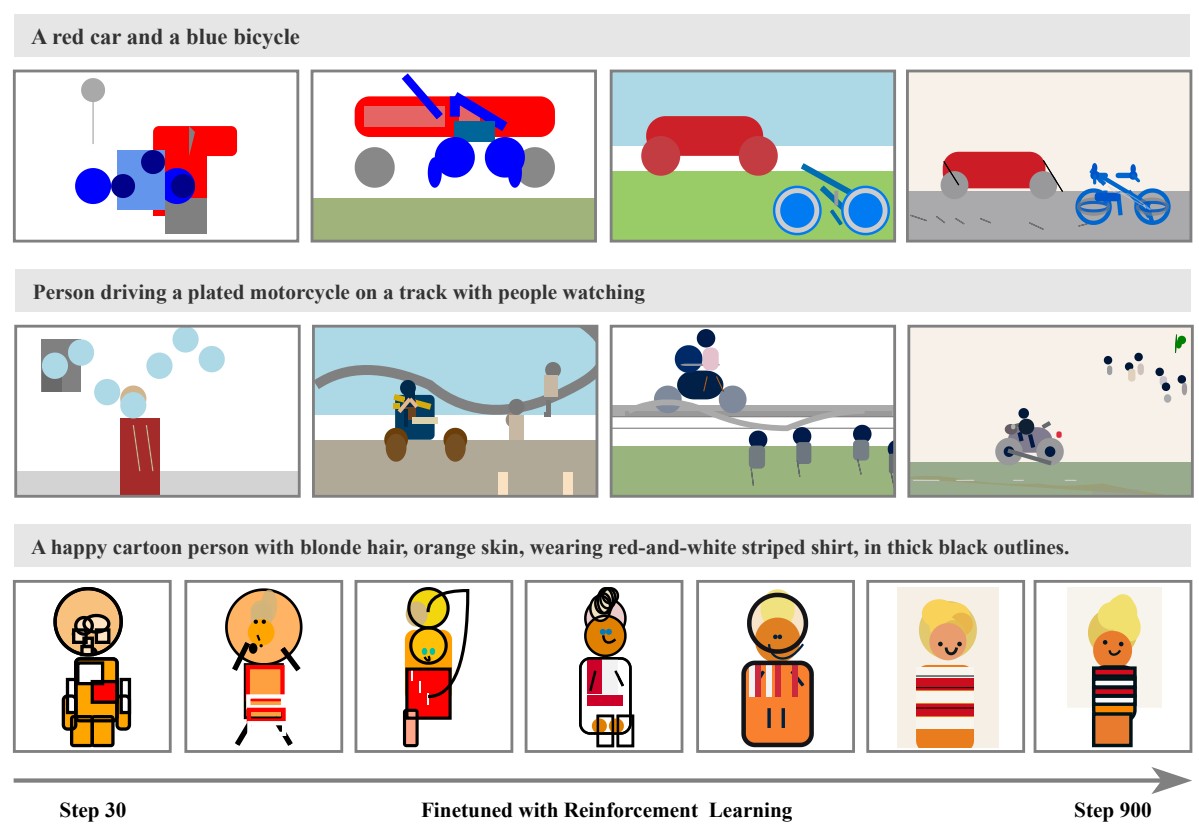

Figure 1: Qualitative results of symbolic graphics programming. We use reinforcement learning with customized verifiable reward to finetune Qwen-2.5-7B. As training progresses, we can observe that the model acquires better compositional drawing ability, producing semantically accurate symbolic graphics programs.

Scalable Vector Graphics (SVGs) are a representative form of SGPs and are widely available on the internet. As renderable programs, they bridge the visual and linguistic domains, framing SGP generation as a semantic grounding task from natural language prompts to formal code. Because the outputs of SVGs can be directly visualized for semantic correctness, SVGs provide an ideal testbed for studying symbolic graphics programming with LLMs. In this work, we restrict the format of SGPs to SVGs, though our methodology can naturally extend to other SGP formats.

We start with the first research question: To what extent can LLMs generate SGPs effectively? To investigate this, we introduce SGP-GenBench, a large-scale benchmark designed to evaluate LLM performance across three dimensions: object-level accuracy, scene-level semantics, and compositional consistency. SGP-GenBench enables systematic comparison across models and provides diagnostic insights into their symbolic graphics programming capabilities. Using this benchmark, we conduct extensive evaluations of both proprietary and open-source models. Our results indicate that proprietary reasoning-enhanced models consistently outperform non-reasoning variants, with performance strongly correlated with coding proficiency. Additionally, open-source LLMs, in contrast to proprietary ones, remain substantially less effective in generating valid and semantically aligned SVGs.

The observation that open-source models still struggle to generate usable SGPs raises the second research question: How can we improve their SGP generation ability? To this end, we propose a reinforcement learning (RL) approach that leverages similarity scores between visual encoder outputs and input text descriptions as the verifiable reward signals. This approach enables LLMs to progressively improve both the quality and semantic alignment of their SVG generation. Experiments show that our method can substantially enhance symbolic graphics programming: open-source LLMs that initially produced unrecognizable SVGs are trained to achieve performance comparable to state-of-the-art proprietary models.

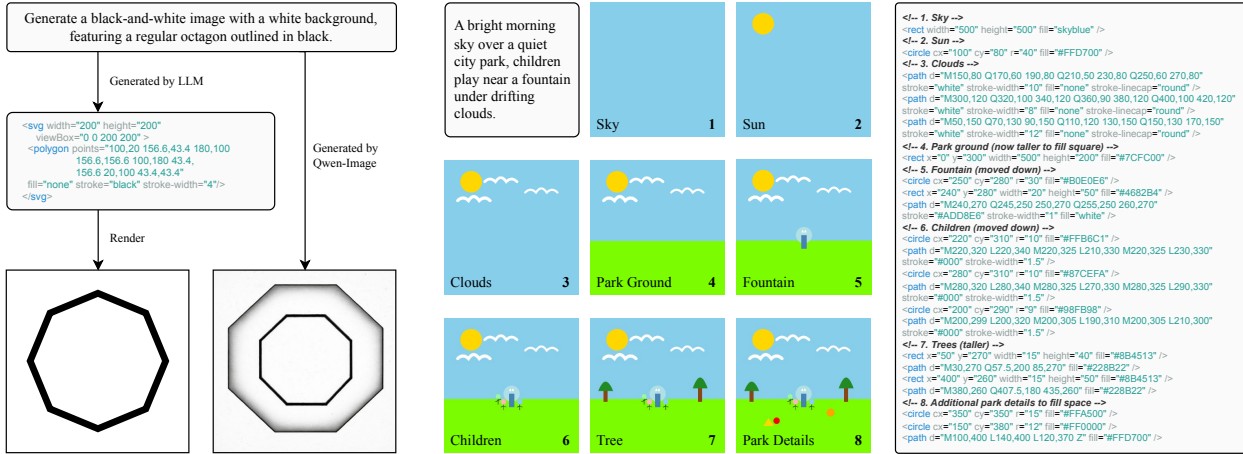

(a) SGPs are driven by symbolic parameters     (b) SGPs are procedural, enabling layer-wise semantic composition

Figure 2: (a) Symbolic controllability of SGPs: to generate a "regular octagon", SGPs can deliver precise representation and fine-grained controllability, in contrast to the result from Qwen-Image (Wu et al., 2025a). (b) Procedural generation: this example illustrates the procedural generation of a park scene, where items are represented at different steps.

In addition, we analyze the results of the RL-trained model to better understand how its SVG generation evolves during training. Through comprehensive case studies and statistical analysis, we identify two key behaviors: (1) the model learns to generate longer and higher-quality SGPs by decomposing complex objects into simpler, more controllable elements, and (2) it produces additional, semantically related visual subjects that well align with the prompt. These findings suggest that our RL-trained model exhibits emergent behaviors beyond those directly optimized for. Our contributions are summarized below:

- We introduce SGP-GenBench, a large-scale benchmark that comprehensively evaluates LLMs' ability to generate SGPs across object-level accuracy, scene-level coherence, and compositional consistency.

- We enhance LLMs' symbolic graphics programming with rule-based reinforcement learning, leveraging similarity between visual encoder outputs and input text descriptions as the verifiable reward.

- We provide an in-depth analysis of RL-trained models, showing that they exhibit emergent behaviors such as decomposing complex concepts into simpler elements and generating additional, semantically relevant objects.

## 2 Symbolic Graphics Programming as Visual Synthesis

Symbolic graphics programming is the task of generating symbolic graphics programs from natural language instructions. Since a symbolic graphics program can be deterministically rendered into a unique graphical object (*e.g.*, an SVG rendered as an image), the task can be regarded as a form of visual synthesis. However, unlike conventional text-to-image generation, which relies on latent representations and pixel-based output, symbolic graphics programming operates by translating natural language into a formal language (*i.e.*, from prompt to code). This distinction highlights its unique nature: visual generation through structured, interpretable, symbolic representations rather than through latent embeddings.

As one of the most widely used visual representations on the internet, SVG serves as a natural bridge between vision and language, making SVG generation a semantic grounding task from prompt to code. The ability to render outputs provides an immediate means of verifying whether the generated program produces the intended result. For these reasons, we adopt SVG as the target representation in the experiments and analyses that follow. The reason why symbolic graphics programming is interesting can be understood through two defining properties of SGPs:

**SGPs are parametric, enabling precise expression.** A symbolic graphics program is inherently parametric: geometry is defined by numeric coordinates, control points, radii, angles, and affine transforms, while appearance is governed by discrete attributes and continuous values (*e.g.*, stroke width, opacity). This parameterization provides precise and scalable control over positions, sizes, alignments, symmetries, and occlusions, allowing models to specify not only what to draw but also how to draw it with fine-grained accuracy. Since exact expression depends on precise parameter values, large language models that yield strong symbolic reasoning can generate accurate geometric graphics, which is a capability that remains difficult for many text-to-image (T2I) systems. As illustrated in Figure 2(a), prompting an LLM and a state-of-the-art T2I model (Qwen-Image, Wu et al. (2025a)) to produce a regular octagon showed a clear difference: the LLM generated SVG with correct vertex coordinates that rendered exactly as intended, whereas the T2I model failed to produce a clean and text-aligned polygon.

**SGPs are procedural, enabling hierarchical semantic composition.** A symbolic graphics program also provides a procedural description, whereby complex scenes are constructed from predefined primitives. In formats such as SVG, this procedural nature is expressed through hierarchical rendering, where later elements occlude earlier ones. This design enables distinct visual concepts to be assigned to separate layers, facilitating operations such as adding, removing, or duplicating elements, as well as reordering them, without disrupting the semantics encoded in individual components. As shown in Figure 2(b), we illustrate a city-park scene by cumulatively stacking elements layer by layer, demonstrating SVG's strong compositional flexibility.

More broadly, the complexity of the generated SGPs (*e.g.*, program length, number of primitives, or nesting depth) can serve as a characterization of the visual complexity underlying a natural language scene description. Intuitively, simple prompts often map to concise programs with few elements, while richer and more detailed descriptions require longer programs with multiple objects, relations, and layered attributes. This provides a structured and quantifiable lens to analyze scene complexity: program statistics can be directly correlated with the semantic richness of the input text and the perceptual intricacy of the target image. Beyond evaluation, such complexity measures can be used to guide curriculum learning, assess model scalability, or even benchmark the compositional reasoning ability of LLMs.

## 3 SGP-GenBench: A Large-Scale Benchmark for Symbolic Graphics Programming

In this section, we introduce SGP-GenBench, a large-scale benchmark designed to evaluate the symbolic graphics programming capabilities of LLMs. The three data components of the benchmark are detailed in Section 3.1, and the evaluation metrics are described in Section 3.2. Figure 3 gives an overview and some examples of our SGP-GenBench.

### 3.1 Construction of SGP-GenBench

We present SGP-GenBench, consisting of three complementary components to comprehensively evaluate and benchmark the SGP generation capabilities of large language models:

- **Scene generation capability** on COCO-VAL, which contains 80 diverse object categories with rich descriptive captions depicting complex scenes with multiple objects and interactions, serving as the comprehensive scene component of our SGP-GenBench. The original validation set from the official 2017 split of MS-COCO (Lin et al., 2014) contains 5,000 images, from which we randomly sampled 1,024 examples for our evaluations to ensure computational efficiency while maintaining statistical significance.

- **Object generation capability** on SGP, a validation set comprising 930 examples from our internet-collected SGP-OBJECT-VAL dataset with captions generated by prompting gemini-2.5-flash-preview, primarily focusing on single object generation tasks to evaluate the model's ability to render individual objects with high fidelity.

- **Compositional generation capability** on SGP-COMPBENCH, which evaluates three key compositional aspects inspired by T2I-CompBench (Huang et al., 2023): *attribute binding* (color, shape, texture), *spatial relationships* (2D, 3D, implicit relations), and *numeracy* (accurate generation of 3-10 objects). The benchmark contains 3,200 prompts for comprehensive evaluation. To ensure models are tested on

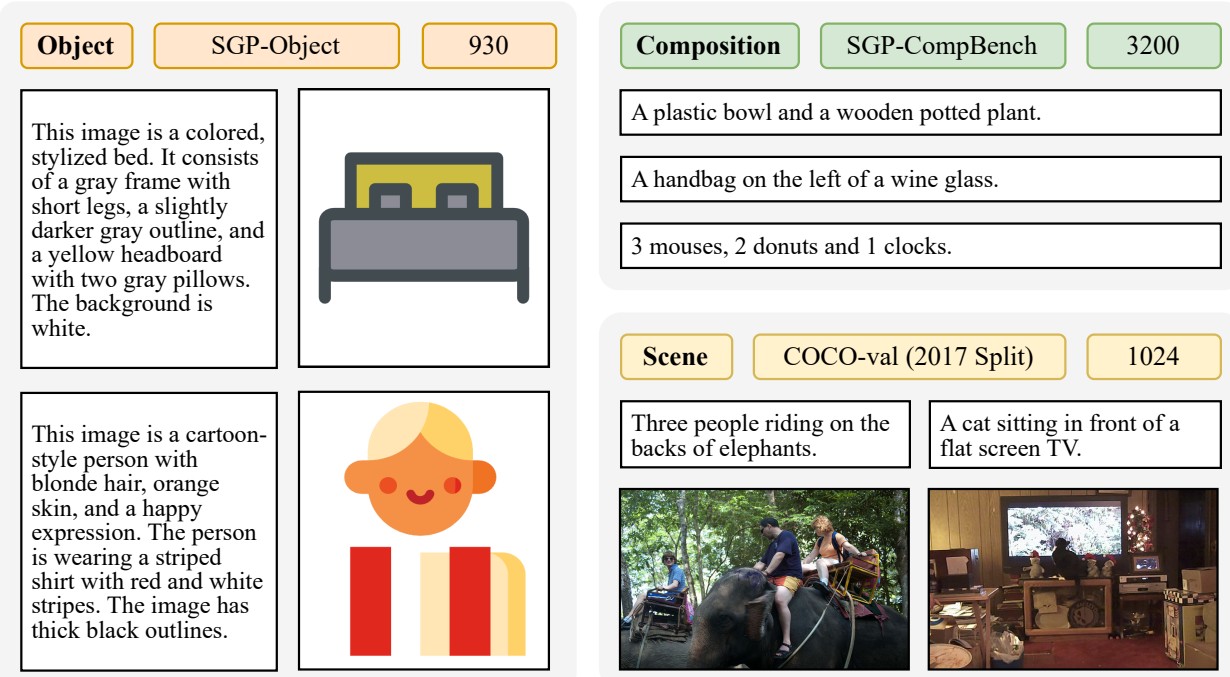

Figure 3: Overview of the proposed SGP-GenBench and some examples.

compositionality of generation rather than object quality of generation, we used 80 common objects as item candidates in our prompt generation. See Table 6 for a comprehensive list of the objects.

## 3.2 Evaluation Metrics

We adopt two categories of evaluation metrics. The first assesses the semantic fidelity of objects and scenes, while the second evaluates the compositional quality of the generated outputs.

To evaluate the semantic fidelity of objects and scenes, we report *CLIP-Score* (Radford et al., 2021; Zhai et al., 2023) (cross-modal cosine similarity between caption and image embeddings), *DINO-Score* (Oquab et al., 2024) (cosine similarity of visual features to a reference) and *VQA-Score* (Hu et al., 2023; Li et al., 2022) (visual question answering accuracy on generated rasters), *HPS v2* (Wu et al., 2023b) (predicted human preference). For a detailed introduction of each metric, refer to Appendix B.2.

For compositional quality, we design prompts for each task and ask the judge model to assess the compositionality of the model; the full prompts are listed in Appendix B.1. Each sub-task is scored out of 100 by asking a judge model whether the generated SVG meets the prompt along a specific dimension. Evaluation prompts for attribute bindings (color, shape, texture) and spacial relations (2D, 3D, implicit) are direct. For numeracy, we assess generation quality in three ways: accuracy of the total number (total), recognizability of all items (item), and correctness of the count for each distinct item (CPI: count per item). The overall numeracy score is the weighted sum of these three components, using weights 0.2, 0.2, and 0.6, respectively.

## 3.3 Summary of Benchmark Results

We summarize the main findings from SGP-GenBench. The benchmark results are shown in two tables (Tables 1 and 2), and complete results along with a more detailed analysis are provided in Section 5.2.

- **SGP-GenBench reflects general model abilities.** The ranking of models on our benchmark aligns well with their perceived general capabilities, especially in code generation. For example, Claude 3.7 Sonnet Thinking generally outperforms o3, which in turn surpasses Gemini 2.5 Pro Preview, followed

by open-source systems like DeepSeek-R1 and Qwen-2.5-7B. This consistent ordering suggests that SVG generation is a reliable indicator of broader model competence.

- **Closed-source models remain strongest.** Frontier systems achieve the best results not only on compositional reasoning tasks such as attribute binding and numeracy, where Claude 3.7 Sonnet Thinking reaches 90.5 on color binding and 89.4 on numeracy, but also on scene and object fidelity, where Gemini 2.5 Pro Preview attains the top DINO object score of 0.653 and strong VQA scene performance of 0.554.

- **Our RL-trained model substantially narrows the gap.** The RL post-trained Qwen-2.5-7B raises its overall compositional score from 8.8 to 60.8, outperforming all other open-source counterparts such as DeepSeek-R1 and QwQ-32B. It also achieves the best VQA score across all models at 0.596, slightly higher than Claude 3.7 Sonnet Thinking, demonstrating that reinforcement learning enables open-source models to approach the closed-source frontier.

## 4 Eliciting Symbolic Graphic Programming from LLMs via Reinforcement Learning with Cross-Modality Alignment Reward

We introduce our problem formulation and reward design in this section, with a schematic illustration of the method in Figure 4. Implementation details (tricks for stabilizing training and preventing reward hacking) are deferred to Appendix D.

### 4.1 Problem Formulation

We start by formulating the symbolic graphics programming task as a rule-based RL problem.

**Task.** Let $\mathcal{C}$ denote captions and $\mathcal{S}$ denote valid SVG programs $s = (s_1, \ldots, s_T)$ of length $T$ over a vocabulary $V$. For a caption $c \in \mathcal{C}$ we draw $s \sim \pi_\theta(\cdot \mid c)$, render it with a deterministic renderer $\hat{\mathbf{x}} = \mathcal{R}(s) \in \mathbb{R}^{H \times W \times 3}$. We cast generation as a single-episode Markov decision process whose state at step $t$ is the pair $(c, s_{1:t-1})$; the action is the next token $s_t \in V \cup \texttt{<eos>}$; the transition deterministically appends $s_t$; the process terminates when $\texttt{<eos>}$ is emitted or the sequence reaches length $T_{\max}$; and a scalar reward $r(s, c, \mathbf{x})$ is issued once upon termination.

**Objective.** We optimize the policy parameters $\theta$ to maximize the expected reward under the data distribution $\mu$:

$$J(\theta) = \mathbb{E}_{(c,\mathbf{x}) \sim \mu} \mathbb{E}_{s \sim \pi_\theta(\cdot|c)} \big[ r(s, c, \mathbf{x}) \big], \tag{4.1}$$

where each data entry consists of a caption $c$ and optionally a reference image $\mathbf{x}$.

**Policy update (GRPO).** We adopt GRPO (Shao et al., 2024), a critic-free variant of PPO (Schulman et al., 2017). For each caption we sample SVG programs $\{s_i\}_{i=1}^G$. With clip range $\epsilon$ and reward $R_i = r(s_i, c, \mathbf{x})$,

$$\mathcal{J}_{\text{GRPO}}(\theta) = \mathbb{E}_{(c,x) \sim \mu, \{s_i\}_{i=1}^G \sim \pi_\theta(\cdot|c)} \left[ \frac{1}{G} \sum_{i=1}^G \frac{1}{|s_i|} \sum_{t=1}^{|s_i|} \left( \min \left( r_{i,t}(\theta) \hat{A}_{i,t}, \text{clip}\left(r_{i,t}(\theta), 1 - \varepsilon, 1 + \varepsilon\right) \hat{A}_{i,t} \right) \right) \right]. \tag{4.2}$$

where $\hat{A}_{i,t}$ and $r_{i,t}$ are defined as

$$\hat{A}_{i,t} = \frac{R_i - \text{mean}\left(\{R_i\}_{i=1}^G\right)}{\text{std}\left(\{R_i\}_{i=1}^G\right)}, \quad r_{i,t}(\theta) = \frac{\pi_\theta(s_{i,t} \mid (c, s_{i,1:t-1}))}{\pi_{\theta_{\text{old}}}(s_{i,t} \mid (c, s_{i,1:t-1}))}. \tag{4.3}$$

### 4.2 Reward Design

For every trajectory (caption $c$, generated SVG program $s$) we assign a scalar reward that factorizes into an outer format gate and an inner perceptual term:

$$r(s, c, \mathbf{x}) = r_{\text{fmt}}(s) \left( \lambda_{\text{Text}} \, r_{\text{Text}}(s, c) + \lambda_{\text{Image}} \, r_{\text{Image}}(s, \mathbf{x}) \right). \tag{4.4}$$

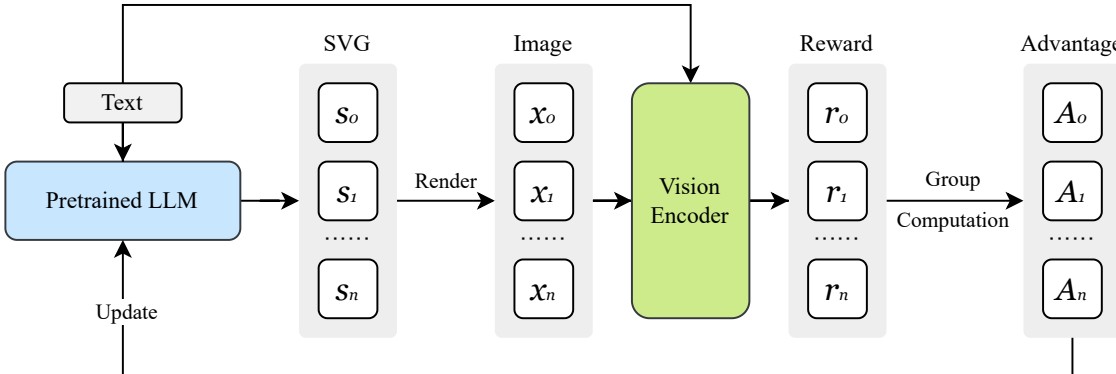

Figure 4: An illustration of the RL pipeline. Given a text description, we sample a group of SVG codes from the model and render them as images. Each SVG code is scored by the alignment between the rendered image and the text description. The advantages are calculated based on the scores, and used for updating the model.

The binary term $r_{\text{fmt}} \in \{0, 1\}$ (defined below) guarantees that only syntactically valid, renderable code propagates perceptual rewards. Throughout, we keep $\lambda_{\text{Text}} = 1$ unless stated otherwise. When a reference image $\mathbf{x}$ is unavailable, we set $\lambda_{\text{Image}} = 0$ without changing Equation (4.4).

### 4.2.1  Format–Validity Reward

We set format-validity reward as a binary reward:

$$r_{\text{fmt}}(s) = \begin{cases} 1, & \text{if } s \text{ passes both checks below,} \\ 0, & \text{otherwise.} \end{cases} \tag{4.5}$$

- **"Think–Answer" structure check.** Each LLM response must follow the prompt template `<THINK>`... `</THINK><ANSWER>`... `</ANSWER>`, where the `ANSWER` block contains SVG code and the `THINK` block contains reasoning process. We apply a lightweight regular expression to verify the presence and order of these tags.
- **Renderer check.** The extracted SVG string is rendered through CairoSVG in `Python`. Successful conversion to a raster image without exceptions constitutes a pass.

### 4.2.2  Text-Image Alignment Reward

We employ a generic language-image contrastive model $\mathcal{E} = (f_{\text{text}}, f_{\text{img}})$, such as CLIP (Radford et al., 2021), SigLIP (Zhai et al., 2023), or any successor trained with a contrastive objective. Given the caption $c$ and the rendered image $\hat{\mathbf{x}} = \mathcal{R}(s)$, we obtain unit-normalized embeddings

$$\mathbf{t} = \frac{f_{\text{text}}(c)}{\|f_{\text{text}}(c)\|_2}, \qquad \mathbf{v} = \frac{f_{\text{img}}(\hat{\mathbf{x}})}{\|f_{\text{img}}(\hat{\mathbf{x}})\|_2}. \tag{4.6}$$

The raw similarity $\cos(\mathbf{t}, \mathbf{v}) = \mathbf{t}^\top \mathbf{v} \in [-1, 1]$ is linearly rescaled to the interval $[0, 1]$:

$$r_{\text{Text}}(s, c) = \tfrac{1}{2}\big(\cos(\mathbf{t}, \mathbf{v}) + 1\big). \tag{4.7}$$

No additional learnable parameters are introduced, maintaining the simplicity of the signal. Because it relies solely on the caption and the model-generated image, $r_{\text{Text}}$ is defined for all prompts, serving as the main supervisory signal in open domain settings where no reference image is provided.

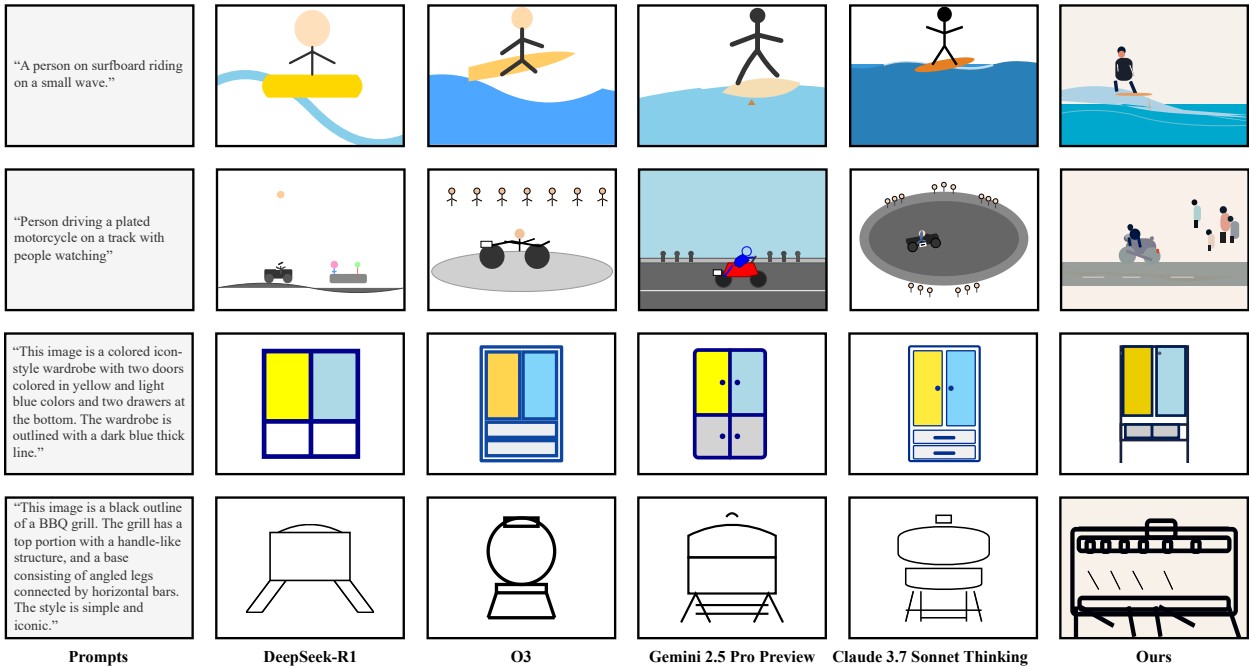

Figure 5: Qualitative comparison of SVGs generated by frontier LLMs and our RL-trained model. Our model achieves results comparable in quality to state-of-the-art commercial models, while generating graphics that are more natural and detailed.

### 4.2.3 Image-Image Alignment Reward

The image-image term $r_{\text{Image}}$ is computed only when a reference image $\mathbf{x}$ is available alongside the caption. We extract global (image-level) features with a self-supervised vision encoder $\mathcal{F}_{\text{DINO}}$ (Oquab et al., 2024):

$$\mathbf{z}_{\text{gt}} = \mathcal{F}_{\text{DINO}}(\mathbf{x}), \qquad \mathbf{z}_{\text{gen}} = \mathcal{F}_{\text{DINO}}(\hat{\mathbf{x}}). \tag{4.8}$$

Then the similarity is measured by the cosine of the two embeddings and then linearly scaled to $[0, 1]$:

$$r_{\text{Image}}(s, \mathbf{x}) = \frac{1}{2}\big(\cos(\mathbf{z}_{\text{gen}}, \mathbf{z}_{\text{gt}}) + 1\big), \tag{4.9}$$

which is used as the visual-fidelity component of the overall reward.

### 4.3 Discussions and Intriguing Insights

In this section, we discuss why applying reinforcement learning with verifiable rewards to symbolic graphics programming is conceptually compelling.

**Distilling knowledge from vision foundation models**. Our reward function is defined by external vision foundation models (*e.g.*, DINO, CLIP), which provide strong semantic and geometric supervision signals. Through reinforcement learning, the LLM gradually aligns its generations with the representations and judgments of these vision models. This process can be viewed as a form of *implicit distillation*, where the LLM internalizes the visual priors, spatial reasoning, and semantic grounding capabilities embedded in large vision models. Beyond improving native visual understanding, such distillation enhances cross-modal alignment, enabling the LLM to better reason about text-SGP-image correspondences, capture fine-grained visual semantics, and ground abstract descriptions in concrete procedural structures.

**Training without ground truth SGPs**. Unlike supervised finetuning (SFT), our RL approach does not require paired image-program annotations or ground truth SGPs. Instead, it can operate directly on raw images, using external vision foundation models to provide feedback signals. This removes the costly and

| Model | CLIP ↑ | | | DINO ↑ | | | VQA ↑ | | | HPS ↑ | | |
|---|---|---|---|---|---|---|---|---|---|---|---|---|
| | Sce. | Obj. | Avg. | Sce. | Obj. | Avg. | Sce. | Obj. | Avg. | Sce. | Obj. | Avg. |
| *Frontier closed-source LLMs* | | | | | | | | | | | | |
| GPT-4o-mini | 0.207 | 0.278 | 0.243 | 0.021 | 0.573 | 0.297 | 0.295 | 0.465 | 0.380 | 0.118 | 0.174 | 0.146 |
| GPT-4o | 0.219 | 0.284 | 0.252 | 0.031 | 0.602 | 0.316 | 0.338 | 0.497 | 0.417 | 0.125 | 0.182 | 0.153 |
| o1-mini | 0.221 | 0.289 | 0.255 | 0.028 | 0.603 | 0.315 | 0.330 | 0.508 | 0.419 | 0.121 | 0.185 | 0.153 |
| o1 | 0.220 | 0.285 | 0.252 | 0.031 | 0.607 | 0.319 | 0.354 | 0.520 | 0.437 | 0.122 | 0.185 | 0.153 |
| o3-mini | 0.231 | 0.293 | 0.262 | 0.036 | 0.608 | 0.322 | 0.379 | 0.520 | 0.450 | 0.128 | 0.187 | 0.158 |
| o3 | 0.253 | 0.283 | 0.268 | 0.067 | 0.595 | 0.331 | 0.521 | 0.482 | 0.502 | 0.153 | 0.180 | 0.166 |
| o4-mini | 0.246 | 0.296 | 0.271 | 0.052 | 0.629 | 0.340 | 0.469 | 0.536 | 0.503 | 0.143 | 0.193 | 0.168 |
| Gemini 2.0 Flash | 0.204 | 0.275 | 0.240 | 0.023 | 0.588 | 0.306 | 0.293 | 0.468 | 0.381 | 0.116 | 0.176 | 0.146 |
| Gemini 2.5 Flash Preview | 0.222 | 0.286 | 0.254 | 0.033 | 0.603 | 0.318 | 0.349 | 0.498 | 0.424 | 0.125 | 0.183 | 0.154 |
| Gemini 2.5 Pro Preview | 0.256 | 0.302 | 0.279 | 0.088 | **0.653** | **0.371** | 0.554 | 0.572 | 0.563 | 0.154 | 0.199 | 0.177 |
| Claude 3.5 Sonnet | 0.240 | 0.293 | 0.266 | 0.055 | 0.624 | 0.340 | 0.428 | 0.528 | 0.478 | 0.140 | 0.190 | 0.165 |
| Claude 3.7 Sonnet | **0.262** | **0.306** | **0.284** | 0.088 | 0.647 | 0.368 | 0.581 | 0.567 | 0.574 | **0.165** | **0.200** | **0.183** |
| Claude 3.7 Thinking | **0.262** | 0.305 | **0.284** | **0.090** | 0.642 | 0.366 | **0.594** | **0.574** | **0.584** | 0.164 | 0.199 | 0.181 |
| *Open-source LLMs* | | | | | | | | | | | | |
| QwQ-32B | 0.219 | 0.272 | 0.245 | 0.031 | 0.549 | 0.290 | 0.334 | 0.456 | 0.395 | 0.123 | 0.172 | 0.147 |
| DeepSeek-R1 | 0.228 | 0.278 | 0.253 | 0.042 | **0.594** | 0.318 | 0.416 | 0.508 | 0.462 | 0.134 | **0.180** | 0.157 |
| Qwen-2.5-7B | 0.155 | 0.213 | 0.184 | 0.008 | 0.400 | 0.204 | 0.265 | 0.385 | 0.325 | 0.103 | 0.148 | 0.125 |
| Qwen-2.5-7B w/ RL (ours) | **0.258** | 0.286 | **0.272** | **0.102** | 0.566 | **0.334** | **0.632** | 0.560 | **0.596** | **0.150** | 0.177 | **0.164** |

Table 1: Performance on scene (COCO-VAL) and object (SGP-OBJECT-VAL) generation. Bold marks the best per group. Our RL-tuned model substantially improves over its base (Qwen-2.5-7B) and is competitive with frontier closed-source models.

often infeasible requirement of constructing large-scale datasets of image-program pairs, which are difficult to collect and scale beyond narrow domains. By learning directly from images, the model can generalize to more diverse and open-ended visual inputs, while the reward mechanism ensures that the generated programs remain semantically faithful to the underlying visual content. This paradigm enables scalable training at internet scale where explicit program annotations are unavailable.

**Alignment between linguistic world and visual world**. Our RL approach, guided by rewards from vision foundation models, can be viewed as an *effective alignment* between the LLM's linguistic understanding (expressed through symbolic graphics programs, SGPs) and the visual knowledge embedded in powerful vision models. In this process, the LLM learns not only to map natural language into structured symbolic programs, but also to ensure that these programs are consistent with the perceptual judgments of vision models. This dual alignment anchors abstract linguistic semantics to concrete visual evidence, narrowing the gap between how language describes a scene and how vision perceives it. Such grounding improves the reliability of the generated SGPs and enhances cross-modal reasoning.

# 5 Experiments and Results

## 5.1 Training Data

We train on a balanced mixture of two sources, COCO 2017 captions and MMSVG-Illustration-40k, yielding 95,026 training examples. From the official MS-COCO 2017 split (118,287 images, each with five human-written captions), we randomly sample 47,513 image-caption pairs. From the MMSVG-2M-Illustration corpus (Yang et al., 2025), we select 47,513 SVGs after filtering out text-centric items (e.g., "letter," "Chinese character," "text").

COCO provides broad, human-authored captions of real-world images, while MMSVG provides detailed captions of vector-graphics. The mixture of the two datasets balances between scene-level and object-level captions, and removing text-related images avoids shortcuts via text rendering.

| Model | Attribute Binding ↑ | | | | Spatial Relation↑ | | | | Numeracy ↑ | | | | Avg. ↑ |
|---|---|---|---|---|---|---|---|---|---|---|---|---|---|
| | Color | Shape | Texture | Avg. | 2D | 3D | Implicit | Avg. | Total | Item | CPI | Overall | |
| *Frontier closed-source LLMs* | | | | | | | | | | | | | |
| GPT-4o | 62.2 | 48.7 | 34.3 | 48.4 | 49.7 | 37.3 | 49.2 | 45.4 | 85.9 | 25.5 | 51.1 | 52.7 | 48.3 |
| o1 | 70.8 | 25.2 | 53.0 | 49.6 | 54.6 | 39.4 | 46.4 | 46.8 | 66.4 | 20.1 | 41.7 | 42.0 | 46.7 |
| o3 | 88.9 | 73.6 | 71.7 | 78.1 | **81.6** | 62.0 | 84.5 | 76.0 | 91.6 | 59.8 | 81.1 | 78.8 | 77.5 |
| o4-mini | 82.4 | 62.1 | 69.6 | 71.4 | 71.0 | 57.9 | 76.5 | 68.5 | 90.3 | 52.9 | 76.1 | 74.3 | 71.0 |
| Gemini 2.5 Flash Preview | 63.6 | 45.0 | 56.9 | 55.2 | 46.0 | 38.9 | 57.1 | 47.3 | 82.8 | 34.5 | 62.0 | 59.8 | 53.4 |
| Gemini 2.5 Pro Preview | 88.1 | 65.7 | 74.9 | 76.2 | 77.4 | 59.1 | 80.0 | 72.2 | 94.7 | 68.0 | 83.8 | 82.3 | 76.2 |
| Claude 3.7 Sonnet | 89.3 | 82.8 | 77.3 | 83.1 | 75.9 | 59.4 | 73.7 | 69.7 | 91.4 | 65.5 | 85.5 | 82.5 | 77.9 |
| Claude 3.7 Sonnet Thinking | **90.5** | **85.6** | **82.4** | **86.2** | 80.2 | **74.4** | **86.4** | **80.3** | **94.9** | **78.9** | **91.4** | **89.4** | **84.8** |
| *Open-source LLMs* | | | | | | | | | | | | | |
| QwQ-32B | 54.3 | 51.0 | 31.4 | 45.6 | 43.6 | 33.5 | 46.0 | 41.0 | 79.9 | 21.1 | 51.4 | 50.9 | 45.2 |
| DeepSeek-R1 | 72.6 | 62.7 | **48.4** | 61.2 | **59.3** | 43.8 | 58.2 | 53.7 | **83.5** | 35.4 | **60.4** | **57.4** | 57.4 |
| Qwen-2.5-7B | 7.1 | 10.0 | 1.7 | 6.3 | 5.2 | 5.8 | 8.1 | 6.4 | 42.6 | 5.8 | 10.7 | 16.1 | 8.8 |
| Qwen-2.5-7B - RL **(Ours)** | **84.3** | **71.3** | 46.0 | **67.2** | 55.7 | **53.9** | **61.7** | **57.1** | 63.4 | **47.5** | 57.6 | 56.8 | **60.8** |

Table 2: Compositional generation results on SGP-CompBench, broken down into attribute binding (color binding, shape binding and texture binding), relation (2D relation, 3D relation and implicit relation), and numeracy (total count, item existence and count per item (CPI)). Average scores are provided for each category and overall. See Table 8 for more performances of more models.

## 5.2 Main Results on SGP-GenBench

We begin by comparing our RL-tuned 7B model with frontier open- and closed-source LLMs on scene and object generation, reporting both quantitative metrics and qualitative examples in Section 5.2.1. We then turn to compositional evaluation on SGP-CompBench in Section 5.2.2. Together, these evaluations provide a complete picture of both fidelity and structured reasoning in SVG generation.

### 5.2.1 Fidelity on Object and Scene Generation

Table 1 compares our model against frontier closed- and open-source LLMs. We report CLIP-Score, DINO-Score, VQA-Score, and HPS v2. Claude 3.7 Sonnet and Thinking leads CLIP and HPS, while Gemini 2.5 Pro Preview tops DINO; however, our **Qwen-2.5-7B w/ RL** attains the best overall VQA score 0.596, exceeding all models. RL lifts the 7B base strongly across metrics, pushing it into the frontier performance band. Across splits, objects generally score higher than scenes on CLIP and HPS, and DINO on scene generation remains low for all due to the photo-vector domain gap. Overall, RL closes much of the gap to proprietary models, yielding superior task-grounded faithfulness while keeping aesthetic quality competitive.

For a qualitative evaluation, we compare our model with other frontier LLMs by examining SVGs generated from four prompts selected from COCO-val and SGP-Object-val, as shown in Figure 5. The results demonstrate that our model usually produce images with enhanced detail fidelity for scene generation. For instance, in response to the prompt `"A person on surfboard riding on a small wave"`, while competing models render only the basic elements (water, surfboard, and athlete), our model incorporates an additional light blue wave layer that accurately represents the foam characteristic of actual surfing conditions. Similarly, for the motorcycle example, our model generates significantly more detailed components, including distinctive red elements representing the `"tail lights"`.

### 5.2.2 Compositional Generation

Following the T2I CompBench (Huang et al., 2025) protocol, in Table 2, we evaluate the compositional capabilities of our model using the SGP-CompBench benchmark. This evaluation is divided into three main aspects: attribute binding (including color, shape, and texture), spatial relationship (covering 2D, 3D, and implicit relations), and numeracy (assessing the model's ability to generate images with object counts ranging from 3 to 10). Detailed evaluation setup can be found in Appendix B.1.

| Encoder | VQA ↑ | | | Diversity ↑ | | | HPS ↑ | | |
|---|---|---|---|---|---|---|---|---|---|
| | COCO | SGP | Avg. | COCO | SGP | Avg. | COCO | SGP | Avg. |
| *Text-Image Encoders* | | | | | | | | | |
| CLIP ViT-B/32 | 0.535 | 0.554 | 0.545 | 0.110 | 0.157 | 0.134 | 0.157 | 0.187 | 0.172 |
| CLIP ViT-L/14 | 0.575 | **0.567** | 0.571 | 0.135 | 0.178 | 0.157 | 0.155 | 0.182 | 0.168 |
| SigLIP Base/16-384 | **0.632** | 0.560 | 0.596 | 0.184 | **0.194** | 0.189 | 0.164 | 0.185 | 0.175 |
| SigLIP Large/16-384 | 0.628 | 0.549 | 0.589 | **0.212** | 0.193 | **0.203** | 0.150 | 0.177 | 0.164 |
| *Vision-Only Encoders (with SigLIP Base/16-384)* | | | | | | | | | |
| w/o Vision Encoder | **0.632** | 0.560 | 0.596 | 0.184 | **0.194** | 0.189 | 0.164 | 0.185 | 0.175 |
| DINOv2-ViT-S/14 | 0.630 | 0.559 | 0.595 | 0.208 | 0.165 | 0.187 | 0.168 | 0.182 | 0.175 |
| DINOv2-ViT-B/14 | **0.632** | 0.558 | 0.595 | 0.174 | 0.139 | 0.157 | **0.173** | 0.182 | **0.178** |
| DINOv2-ViT-L/14 | **0.632** | 0.561 | **0.597** | 0.176 | 0.145 | 0.161 | 0.167 | **0.188** | 0.177 |
| DINOv2-ViT-G/14 | 0.627 | 0.561 | 0.594 | 0.166 | 0.138 | 0.152 | 0.171 | 0.182 | 0.176 |

Table 3: Ablation of embedding models on COCO-val and SGP-Object-val. Bold numbers mark the best results for each column.

**Attribute binding.** All models perform significantly better on color and shape binding compared to texture binding. Our RL-tuned model mirrors this pattern. While narrowing the gap with frontier closed-source models, it performs strongly on color/shape binding but still lags on texture binding, reflecting the inherent difficulty of encoding textures in SVG. This disparity reflects the inherent characteristics of SVG representations. Colors are simple parameters (e.g., *fill="red"*, *stroke="#00FF00"*) and shapes can be controlled through geometric primitives and their attributes, whereas textures such as the highlights and reflections of *a metallic apple* demand extra drawing operations beyond parameter tweaks. Since patterns that cannot be directly assigned through parameters are difficult to express in SVG, the performance drop is therefore expected.

**Spatial relationship.** Models generally handle 2D and implicit relations better than 3D ones. Precisely placing elements suffices for 2D, and implicit cues (e.g., "watch," "wear") tolerate loose layouts. But 3D scenes require ordering code so that later shapes occlude earlier ones, which is harder under SVG's top-to-bottom rendering, hence lower scores.

**Numeracy.** All models demonstrate strong capabilities in overall counts, with Claude 3.7 Thinking achieving an impressive 94.9% accuracy, yet stumble on Count-Per-Item: they may draw seven objects but not the asked "three apples and four bananas." CPI rises alongside general generation quality, implying that recognizable objects are a prerequisite for correct numeracy recognition. We also observe that the Item score fluctuates more than the CPI score. The Item score reflects whether a specific object exists, while the CPI measures the accuracy of the predicted quantity under the assumption that the object is present. Since the CPI explicitly conditions the judge on object existence, it naturally emphasizes numerical accuracy. Thus, the larger swings in Item scores suggest the model still struggles with generating semantically correct objects, whereas the more stable and higher CPI values indicate it can count reliably once recognition is established.

### 5.3 Ablation Studies

To disentangle the factors behind our performance gains, we systematically ablate three core components of the RL pipeline: (i) the reward stack (Section 5.3.1), (ii) the presence of explicit Chain-of-Thought prompting (Section 5.3.2), and (iii) the choice of RL algorithm (Appendix E.1). Each ablation reveals how design choices affect quality and diversity. For the computation details of the diversity, see Appendix B.2.

#### 5.3.1 Effect of Different Embedding Models

Our reward pipeline relies on (i) text-image encoders to score semantic alignment and (ii) vision-only encoders to judge visual similarity (see Section 4.2). To quantify the impact of each choice, we compare the influence of different embedding models. For text-image encoder ablations, we train only with text-image similarity reward, while for vision-only encoder ablations we add both text-image and image-image similarity rewards

| Variant | CLIP ↑ | | | DINO ↑ | | | VQA ↑ | | | HPS ↑ | | | Diversity ↑ | | |
|---------|------|------|------|------|------|------|------|------|------|------|------|------|------|------|------|
| | COCO | SGP | Avg. | COCO | SGP | Avg. | COCO | SGP | Avg. | COCO | SGP | Avg. | COCO | SGP | Avg. |
| w/ CoT | 0.258 | **0.286** | **0.272** | **0.102** | **0.566** | **0.334** | **0.632** | **0.560** | **0.596** | 0.164 | 0.185 | 0.175 | **0.184** | 0.194 | 0.189 |
| w/o CoT | **0.259** | 0.282 | 0.271 | 0.099 | 0.555 | 0.327 | 0.621 | 0.548 | 0.585 | **0.172** | **0.186** | **0.179** | 0.177 | **0.269** | **0.223** |

Table 4: Ablation of Chain-of-Thought prompting on COCO-VAL and SGP-OBJECT-VAL. Performance differences across metrics are marginal, indicating CoT is not essential for quantitative performance.

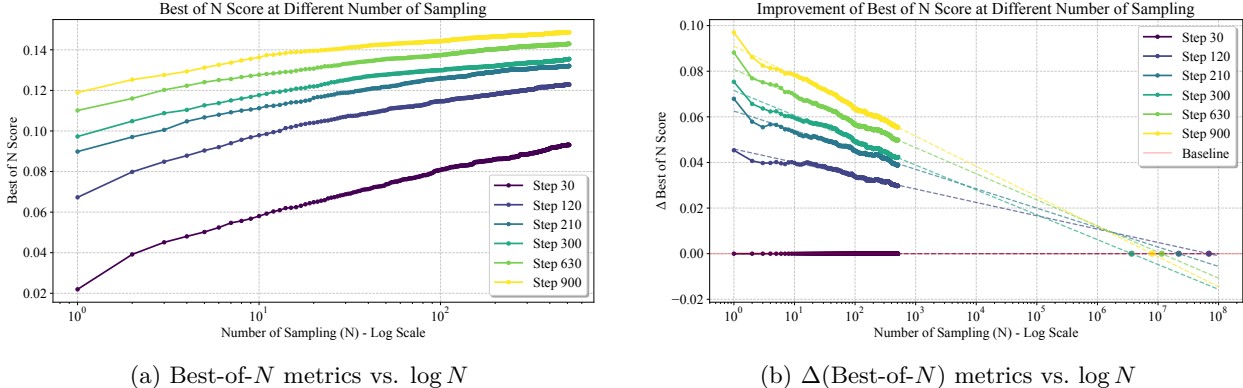

(a) Best-of-$N$ metrics vs. $\log N$          (b) $\Delta$(Best-of-$N$) metrics vs. $\log N$

Figure 6: Analysis of the *Best-of-N* performance of RL checkpoints. (a): *Best-of-N* curves for the SigLIP-Base score. (b): Gain over the first checkpoint; the horizontal intercept indicates the value of $N$ required to match RL-trained later checkpoints. Each curve corresponds to a checkpoint at 30, 120, 210, 300, 630, and 900 RL steps. The $y$-axis shows the text-image similarity computed with SigLIP Base/16-384.

with equal coefficients (1.0) and the text-image encoder is fixed to `SigLIP Base/16-384`. Each setting is trained to step 750 for comparability. As shown in Table 3, three observations emerge from the ablation.

- **SigLIP yields stronger grounding than CLIP.** Replacing CLIP with SigLIP consistently boosts factual grounding (VQA) and diversity of generated results, the gap is especially significant on the natural-image COCO-VAL split but narrows on the synthetic-caption SGP-OBJECT-VAL set, suggesting that SigLIP's training—on more diverse and semantically rich image-text data—better captures real-world semantics, whereas abstract shape-caption alignment benefits less from this advantage.

- **Embedding model size does not strongly correlate with performance.** Within each family, larger encoders do not always guarantee higher VQA: *SigLIP Large/16-384* raises Diversity yet loses a few VQA points relative to the smaller *Base*, and CLIP ViT-L/14 only marginally beats ViT-B/32.

- **Vision-only reward gives little improvements on VQA but improves HPS.** Adding a lightweight *S/14* reward atop SigLIP gives no VQA gain, whereas *L/14* yields a slightly better VQA; the extra-large *G/14* gives diminishing VQA score. In general adding vision-only rewards tends to reduce diversity but improves alignment with human preference. Because additional vision encoders yield only marginal gains, we report all final results using a fixed reward stack of *SigLIP Base/16-384*.

### 5.3.2 Effect of CoT Prompting

We test whether explicit Chain-of-Thought prompting is necessary for performance or primarily beneficial for interpretability. We train the two prompting variants on Qwen2.5-7B for 750 RL steps under an identical reward stack: (1) *With CoT*: model outputs a self-explanation before the SVG; and (2) *Without CoT*: model outputs the SVG directly.

As shown in Table 4, the ablation confirms that explicit CoT prompting is not a prerequisite for strong quantitative performance: across CLIP, DINO, VQA, HPS and Diversity the two variants differ only slightly.

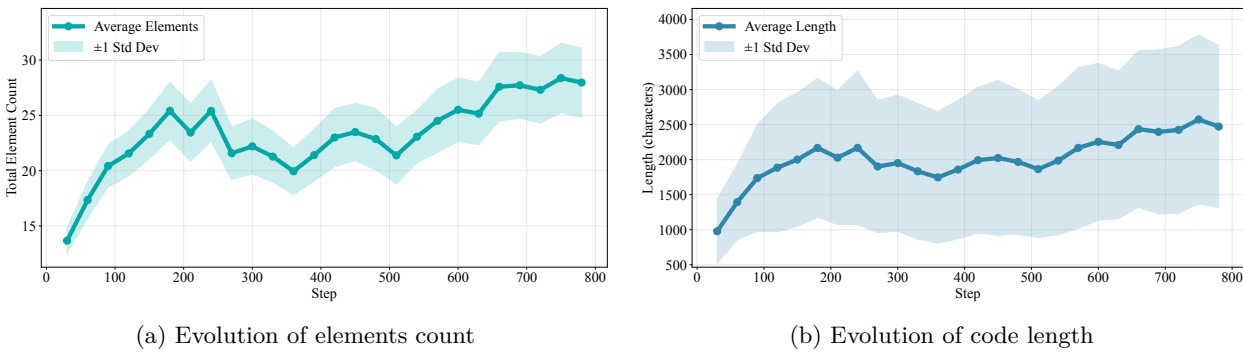

(a) Evolution of elements count

(b) Evolution of code length

Figure 7: Training dynamics of code complexity. Error bars reflect variability across prompts and sampling replicates.

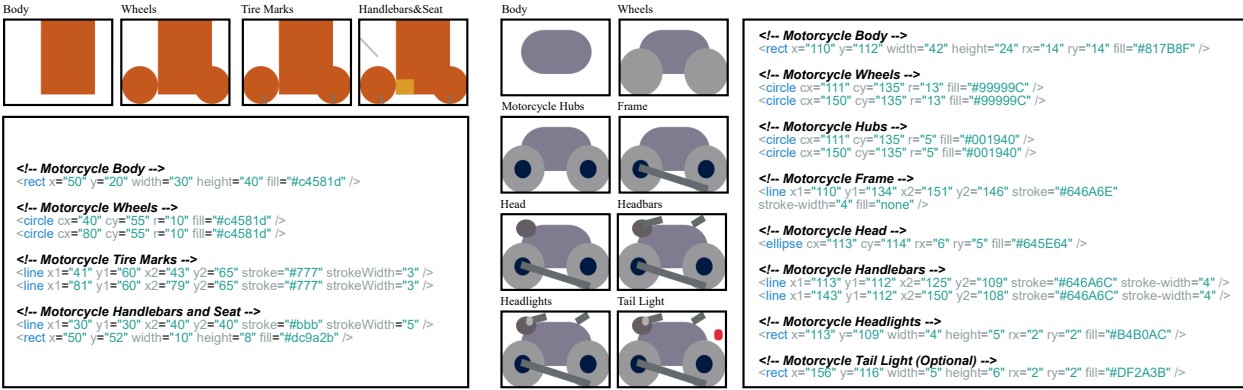

Model-generated motorcycle at step 30

Model-generated motorcycle at step 900

Figure 8: Although both express the concept of "motorcycle," the early-stage model at training step 30 only divides the concept into four levels, with poor semantic representation and inaccurate relative positioning across levels. By step 900, the later model splits the concept into eight parts, achieving accurate semantics and precise positional encoding at every level.

SVG generation itself already induce implicit planning, so the program itself functions as the reasoning trace, and the contribution of verbal reasoning is negligible.

## 6 Additional Analysis of the RL-tuned Model

In this section, we analyze how reinforcement learning with verifiers (RLVR) changes the model's SVG drawing behavior and capability. First, we plot the improvement of *Best-of-N* sampling performance as training progresses in Section 6.1. To better understand the source of this improvement, we analyze the behavioral patterns that emerge during RLVR training in Section 6.2. We discuss an unexpected color-choice bias of different embedding models in Section 6.3.

### 6.1 Reinforcement Learning *vs.* Best-of-N Sampling

Compared to measuring performance by sampling once, *Best-of-N* metrics report the best score among N parallel samples and can better measure the model's potential to complete a task. Yue et al. (2025) has shown that in the math domain the improvement of RL training can be offset by *Best-of-N* sampling with moderate parallel samples N. We thus want to verify if RL can improve the model capability in a non-trivial way. In Figure 6(a) we show that RL shifts the entire *Best-of-N* curve upward for every checkpoint, although

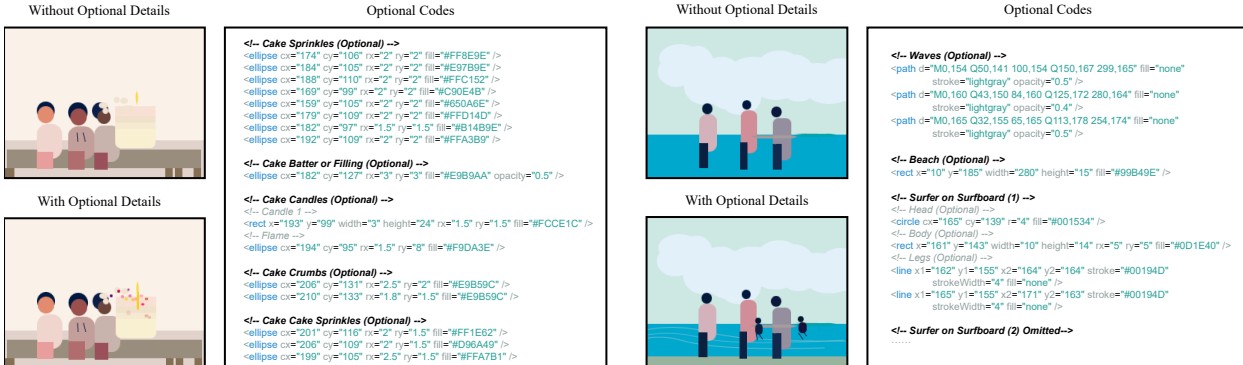

(a) A group of people sit at a table with cake.

(b) People standing in an over cast ski looking out to sea with surf boards.

Figure 9: Examples of optional details generated without explicit prompting. (a) Given only "A group of people sit at a table with cake", the model adds sprinkles on the cake. (b) Given only a beach-related prompt, the model introduces waves, sand, and a surfer. These unrequested elements are consistent with the scene and enhance its naturalness and completeness.

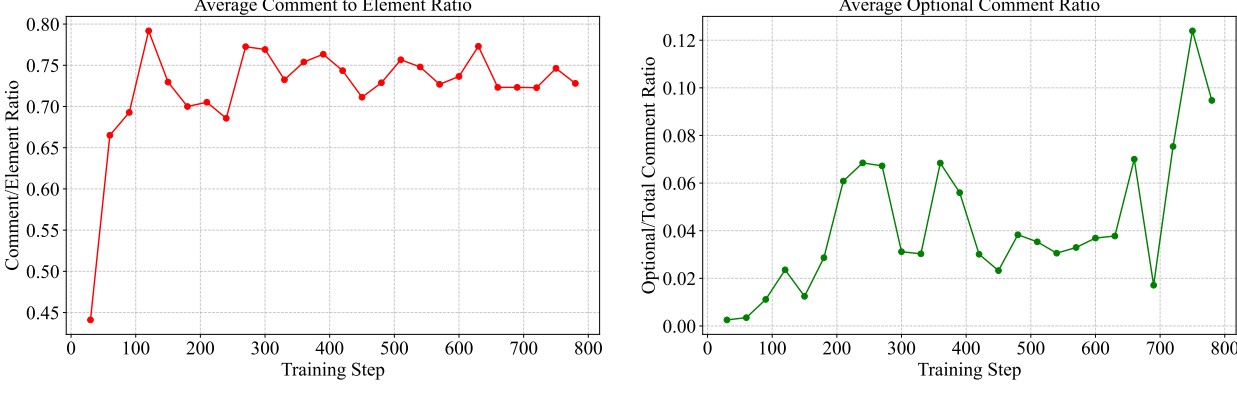

(a) Comment-to-element ratio vs. training step

(b) Optional comment ratio vs. training step

Figure 10: Quantitative evolution of generated code structure over training.

for larger N, the gap shrinks. To determine where curves intersect, we subtract the baseline curve (checkpoint 30) from later curves in Figure 6(b) and linearly fit each curve to locate the $N$ where advantage vanishes. All RL checkpoints intersect with the baseline between $10^6$ and $10^8$ samples, which is three to six orders of magnitude larger than the $10^2$-$10^3$ reported in Yue et al. (2025). This indicates that for the task of symbolic graphics programming, approximately a million candidates would be required to match the benefits RL brings within 1K steps of training, demonstrating that RL delivers qualitative improvements that naive *Best-of-N* decoding cannot realistically achieve within practical compute budgets.

## 6.2 How SGP Generation Capabilities Evolve during Training

Figure 7 shows that both the number of SVG elements and the total code length increase steadily over training. This trend indicates that the policy progressively adopts richer scene structure and longer programs to draw according to the prompts. Closer inspection of the generated code reveals two consistent behaviors that explain these curves.

**Strategy 1: Decomposition into basic components.** The model typically emits a short comment followed by several lines of SVG that implement that sub-concept. Early in training, a single comment often precedes a block of code. As training progresses, the policy decomposes complex objects into multiple

| CLIP reward | | | SigLIP reward | | |
|---|---|---|---|---|---|
| **Caption** | **Image** | **SVG excerpt** | **Caption** | **Image** | **SVG excerpt** |
| Man laughing standing next to his motorcycle with his bicycle attached to it. |  | `<rect x="0" y="200" width="400" height="100" fill="green" stroke="none"/>` | Man laughing standing next to his motorcycle with his bicycle attached to it. |  | `<rect x="10" y="140" width="280" height="45" fill="#A0AAAA"/>` |
| People walking by a blue train next to a mountain. |  | `<rect x="210" y="240" width="30" height="15" fill="royalblue" stroke="black" stroke-width="2"/>` | People walking by a blue train next to a mountain. |  | `<polygon points="30,50 60 ,20 90,50 110,75 130,60 160, 45 190,55 210,70 240, 50 260,80 290,100" fill="#99B09E" />` |

Table 5: Color choices under different reward models. CLIP prefers canonical colors like red and blue, while SigLIP prefers low-saturation colors like #948E8F.

parts that are easier to draw and to place precisely on the canvas, yielding more accurate geometry and spatial relationships (Figure 8). Throughout, the model relies on a limited set of elements to draw(e.g., `rect`, `circle`, `line`, `path`); see Appendix E.3 for distribution of used drawing primitives, suggesting that improved drawings arise from better composition of a fixed toolbox rather than expanding it.

**Strategy 2: Contextual optional details.** Beyond literal prompt fulfillment, the policy increasingly introduces plausible, unrequested elements that improve coherence and realism—e.g., sprinkles on a cake for "people at a table with cake," or waves, sands, and a surfer in beach scenes (Figure 9). These decoration details are consistent with the scene and contribute to perceived completeness.

We quantify these behaviors in Figure 10. The comment-to-element ratio increases with training, reflecting finer-grained decomposition as shown in Figure 10(a), while in Figure 10(b) the fraction of comments annotated "(optional)" rises, indicating more frequent addition of contextual details.

### 6.3 Color Preferences Under Different Reward Models

An unexpected stylistic discrepancy emerges when we compare different text-image encoders. Under a CLIP reward the policy gravitates toward canonical color words—`fill="red"`, `fill="blue"`—whereas the SigLIP reward favors delicate, low-saturation hexadecimal colors such as `fill="#948E8F"`. We place the two behaviors side by side in Table 5.

Qualitatively, SigLIP-rewarded outputs appear less saturated and more diversed. This suggests that SigLIP encourages finer color matching, encouraging the model to move beyond the basic palette.

## 7 Related Work

**Text-to-SVG Generation.** Early pioneering models for vector graphics generation (Carlier et al., 2020; Efimova et al., 2022; Lopes et al., 2019; Reddy et al., 2021) were limited to producing relatively simple or domain-specific images. Additionally, since these approaches typically required curating custom SVG datasets and training models from scratch, they often suffered from poor generalization and did not support text-guided generation. These critical limitations were overcome with the introduction of diffusion-based pipelines, which enabled powerful text-to-SVG generation. Diffusion-based pipelines such as VectorFusion (Jain et al., 2023) and SVGDreamer (Xing et al., 2024) optimize vector primitives by back-propagating pixel-space losses from text-conditioned diffusion models. Moving beyond pure diffusion, LLM-centric approaches (Wu et al., 2023a; Xing et al., 2025b) design SVG-aware encoding: Chat2SVG (Wu et al., 2025b) lets an LLM emit a coarse template that a diffusion stage refines, NeuralSVG (Polaczek et al., 2025) learns an implicit MLP scene representation trained with score-distillation, while StarVector (Rodriguez et al., 2025a) frames SVG code as a sequence in a multi-modal transformer. Reason-SVG (Xing et al., 2025a) employs rule-based verification rewards to enhance text-to-SVG generation fidelity and structure consistency. Similarly, RLVR-driven pipelines have been extended to image-to-SVG generation (Rodriguez et al., 2025b), demonstrating that verifier-guided rewards can effectively capture geometric and perceptual alignment beyond text-only

supervision. These works highlight the growing synergy between reinforcement learning and vector generation, motivating our own RL-based framework that achieves high SVG generation quality with compact models.

**Reinforcement-Learning Post-Training for LLMs.** RLHF (Ziegler et al., 2019; Ouyang et al., 2022) popularized by InstructGPT (Ouyang et al., 2022) finetunes policies with PPO (Schulman et al., 2017) against a reward model fitted to human preferences. RLAIF (Lee et al., 2024; Bai et al., 2022) replaces these labels with AI-generated preferences. A complementary line, RLVR, uses hard verifiers instead of subjective scores: DeepSeek-R1 (Guo et al., 2025) shows that purely rule-based reward from mathematical verifiers elicit strong reasoning. follow-up works apply RLVR in coding (Le et al., 2022; Gorinski et al., 2023; Gehring et al., 2025) and agentic tool-use (Qi et al., 2024; Wei et al., 2025). Our work inherits RLVR and RLAIF's label efficiency with foundation models providing rewards, extending RLVR's simple rule-based scenario to a reward that reflects complex human preference and perception.

**Vision and Multi-modal Foundation Models.** Our rewards are calculated by angular distance of vectors in the embedding space of foundation encoders. DINOv2 (Oquab et al., 2024) extracts rich visual features, letting us judge visual similarity of an SVG render and a reference image. CLIP (Radford et al., 2021) aligns images and text in a shared space, while SigLIP (Zhai et al., 2023) refines CLIP's contrastive objective with a pairwise sigmoid loss. Together these encoders eliminate task-specific labeling and keep our RL loop lightweight.

**Benchmarking LLMs on Vector Graphics Processing.** Evaluating Large Language Models (LLMs) for vector graphics processing is a nascent but growing research area. While several benchmarks have been proposed, they often fall short in assessing the complexity of real-world SVG generation. For instance, BBH (Suzgun et al., 2022) utilizes SVG primitives to assess LLMs' understanding of basic geometric shapes. SGP-Bench (Qiu et al., 2025) focuses on LLMs' ability to semantically understand symbolic graphics programs, using SVG code as an indicator of 2D modality comprehension, but does not evaluate LLM's code generation capabilities. SVGEditBench (Nishina & Matsui, 2024; 2025) proposes a benchmark for assessing LLMs' ability to modify SVG code, though the required modifications remain largely simple. While VGBench (Zou et al., 2024b) broadens the scope by evaluating both LLM's understanding and generation of vector graphics, its SVG generation benchmarks are confined to simple SVG icons, lacking the intricate complexity characteristic of professional vector designs. In contrast, our proposed SGP-GenBench directly addresses this gap by providing a benchmark specifically designed to evaluate LLMs' ability to generate complex vector graphics, offering a significantly more challenging and comprehensive assessment than existing methods.

## 8 Concluding Remarks

In this paper, we address two key questions: (1) What is the current quality of symbolic graphics program generation by large language models? and (2) How can it be improved? To this end, we first introduce SGP-GenBench, a benchmark that evaluates LLMs' ability to generate SGPs along three dimensions: object-level accuracy, scene-level coherence, and compositional consistency. Second, we propose a post-training approach that finetunes models with rule-based reinforcement learning, where the similarity between the rendered image and the input text serves as a verifiable reward. Experimental results demonstrate that the finetuned model generates more accurate and detailed SGPs while also acquiring effective generation strategies. Looking ahead, promising directions include developing adaptive curricula, analyzing the evolution of models' internal processes, and exploring whether enhanced drawing skills can transfer to broader reasoning tasks.

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

# Appendix

## Table of Contents

# A   Data Curation

## A.1   SGP-Object

We first identified the main categories needed for our collection, then determined the appropriate subcategories. We gathered SVG data by crawling categorized content from searchable SVG repositories such as SVGRepo. After deduplication, we used a vision-language model (Gemini-2.0-Flash) to generate captions for each image. Our quality checks confirmed this model was well-suited for the captioning task.

Our collection resulted in 9,268 SVG samples. We reserved 10% for the evaluation set, with some subcategories receiving higher sampling rates to ensure diversity. This yielded 930 samples for evaluation purposes.

The remaining data was initially intended for training. However, considering that MMSVG might already contain similar samples, we decided not to use this data for training. We plan to release this dataset to the research community in the future.

To prevent models from exploiting SVG's text rendering capabilities to directly output text and arti-ficially inflate evaluation scores, we implemented strict filtering mechanisms. Specifically, we removed all SVG samples containing text-related tags such as `<text>`, `<tspan>`, and `<textPath>` from our dataset. Additionally, we filtered out prompts containing words related to text elements, including: 'text', 'word', 'letter', 'character', 'symbol', 'number', 'digit', 'font', 'script', 'write', 'written', 'writing', 'typography', 'label', 'caption', 'title', 'name', 'sign', 'signature', 'logo', 'slogan', 'spell', 'phrase', 'quote', and 'message'. This ensures that models must express content through drawing graphical elements rather than simply rendering text, allowing for more accurate assessment of true drawing capabilities.

---

**Prompt Used for Text Detection**

**Prompt:** Based on the following image description, analyze whether the image likely contains any text, letters, words, numbers, characters, symbols, or other textual elements.
Image Description: "description"
First provide your reasoning about whether the description suggests text is present in the image. Then provide your determination with a simple "Contains text: [Yes/No]" on a new line.
**Example output:** The description mentions "a logo with the company name written below", which clearly indicates there is text in the image. Contains text: Yes
**Another example:** The description only mentions "a red circle with blue background" with no mention of any text, letters, or symbols. Contains text: No

---

## A.2   SGP-CompBench

We followed the methodology of T2ICompBench, utilizing the same relation and binding vocabulary, but substituted the object names with the 80 most frequently occurring object nouns from COCO. The primary difference in our approach lies in the evaluation process. Due to the significant distribution gap between SVG images and natural photographs, we were unable to employ trained BLIP models or detection models to assist with scoring. Instead, we relied exclusively on LLMs as judge models for evaluation purposes.

For our SGP-CompBench evaluation, we employed two complementary approaches to generate test prompts:**(1)LLM-Generated Prompts.** For both binding and relation tasks, we used a large language model to create 600 prompts for each category. This approach ensured linguistic diversity and natural phrasing in our test set.**(2)Template-Generated Prompts.** For numeracy tasks, we utilized code templates to systematically generate prompts covering object counts from 3 to 10. For each count value, we created 100 prompts, resulting in 800 numeracy test cases. This methodical approach allowed us to comprehensively evaluate the model's ability to handle different quantities.to

The combined approach of LLM-generated and template-generated prompts provided a robust and diverse evaluation framework, enabling thorough assessment of our model's compositional generation capabilities across different aspects of sgp generation.

# B  Evaluation Details

## B.1  SGP-CompBench Evaluation Details

This section provides the detailed evaluation protocol and prompt design for SGP-CompBench, including the standardized instructions for SVG generation and the quantitative evaluation criteria.

For the numeracy evaluation, we employ a three-step assessment process: verifying the accuracy of total object count (Total), confirming the presence of all required objects (Item Presence), and validating the correct count per specific item (Count Per Item, CPI). These three metrics are weighted at 0.2, 0.2, and 0.6 respectively to calculate the final numeracy score.

To obtain quantitative results, we adopt the Model-as-a-Judge (MAJ) framework, where a vision–language model is prompted to assess whether the generated image satisfies the requirements specified in the caption for each subtask. For each aspect, we design clear and specific prompts tailored to the corresponding evaluation criterion. The model outputs a score ranging from 0 to 100.

### B.1.1  80 Common Objects List

We list here the 80 common objects in Table 6.

| ID Category | ID Category | ID Category | ID Category | ID Category |
|---|---|---|---|---|
| 1 person | 2 bicycle | 3 car | 4 motorcycle | 5 airplane |
| 6 bus | 7 train | 8 truck | 9 boat | 10 traffic light |
| 11 fire hydrant | 12 stop sign | 13 parking meter | 14 bench | 15 bird |
| 16 cat | 17 dog | 18 horse | 19 sheep | 20 cow |
| 21 elephant | 22 bear | 23 zebra | 24 giraffe | 25 backpack |
| 26 umbrella | 27 handbag | 28 tie | 29 suitcase | 30 frisbee |
| 31 skis | 32 snowboard | 33 sports ball | 34 kite | 35 baseball bat |
| 36 baseball glove | 37 skateboard | 38 surfboard | 39 tennis racket | 40 bottle |
| 41 wine glass | 42 cup | 43 fork | 44 knife | 45 spoon |
| 46 bowl | 47 banana | 48 apple | 49 sandwich | 50 orange |
| 51 broccoli | 52 carrot | 53 hot dog | 54 pizza | 55 donut |
| 56 cake | 57 chair | 58 couch | 59 potted plant | 60 bed |
| 61 dining table | 62 toilet | 63 tv | 64 laptop | 65 mouse |
| 66 remote | 67 keyboard | 68 cell phone | 69 microwave | 70 oven |
| 71 toaster | 72 sink | 73 refrigerator | 74 book | 75 clock |
| 76 vase | 77 scissors | 78 teddy bear | 79 hair drier | 80 toothbrush |

Table 6: The 80 common objects for SGP-CompBench.

### B.1.2  SVG Generation Instruction

For the SVG generation task in SGP-CompBench, the following standardized instruction is used to prompt the model:

> **SVG Generation Instruction**
>
> You are an expert in generating SVG code. Your task is to carefully analyze the description and produce only the corresponding SVG code. Do not generate any images or explanations—output strictly the SVG code that fulfills the following description.
> Description: [description]

### B.1.3 Evaluation Prompts and Scoring Criteria

To quantitatively assess the generated SVGs, we design specific evaluation prompts and scoring rubrics for different aspects: attribute binding, relation, and numeracy. The following are the evaluation prompts and their corresponding scoring criteria. We used Gemini-2.5-Flash-Preview as our judge model used in evaluation on SGP-CompBench, for the superior vision-language understanding capability of the model. We used Gemini-2.5-Flash-Preview as our judge model used in evaluation on SGP-CompBench, for the superior vision-language understanding capability of the model.

---

**Prompt Used for Attribute Binding Evaluation**

**Prompt:** Evaluate whether the image matches the following prompt: [PROMPT]
**Scoring criteria:**

- 100: All items are recognizable and the binding between items and their attributes is correct.

- 50: All items are recognizable, but the binding between items and their attributes is incorrect or unclear.

- 30: Items are not recognizable, but the attribute binding appears correct.

- 0: Items are not recognizable and the binding between items and their attributes is incorrect.

**Response format:**

```
REASONING: [your reasoning]
SCORE: [score]
```

---

**Prompt Used for Relation Evaluation**

**Prompt:** Evaluate whether the image matches the following prompt: [PROMPT]
**Scoring criteria:**

- 100: The items are clear and the relation between items is correct.

- 50: The items are not clear, but the relation between items is correct.

- 30: The items are clear, but the relation between items is incorrect.

- 0: The items are not clear and the relation between items is incorrect.

**Response format:**

```
REASONING: [your reasoning]
SCORE: [score]
```

---

**Prompt Used for Numeracy Evaluation (Total Count)**

**Prompt:** Evaluate whether the image contains exactly [TOTAL_COUNT] distinct items in total (they do not need to be recognizable, but should be clearly individual objects).
**Scoring criteria:**

- 100: All items in the image are clearly individual objects, and the total count is correct.

- 50: All items are clearly individual objects, but the total count is incorrect.

- 30: Some items are clearly individual objects, and the total count is incorrect.

- 0: The items are not clearly individual objects and the total count is incorrect.

**Response format:**

```
REASONING: [your really brief reasoning]
SCORE: [score]
```

**Prompt Used for Numeracy Evaluation (Item Presence)**

**Prompt:** Check whether the image contains the following items: [ITEM LIST].
**Scoring criteria:**

- 100: The image contains all the items listed above.

- 50: The image contains most of the items listed above.

- 30: The image contains some of the items listed above.

- 0: The image does not contain any of the items listed above.

**Response format:**

```
REASONING: [your really brief reasoning]
SCORE: [score]
```

**Prompt Used for Numeracy Evaluation (Count Per Instance)**

**Prompt:** Evaluate whether the image contains exactly [COUNT] distinct [NOUN] in total.
**Scoring criteria:**

- 100: The image contains exactly [COUNT] distinct [NOUN], and they are clearly individual objects.

- 50: The image does not contain all the [COUNT] distinct [NOUN], but the count is close to [COUNT].

- 30: The image does not contain all the [COUNT] distinct [NOUN], but the count is far from [COUNT].

- 0: The image does not contain any of the [COUNT] distinct [NOUN].

**Response format:**

```
REASONING: [your really brief reasoning]
SCORE: [score]
```

### B.2  Metrics Details for Scene and Object Evaluation

- **CLIP-Score.** For each generated image we compute the cosine similarity between its vision embedding and the caption embedding using two CLIP Radford et al. (2021) models (`ViT-B/32`, `ViT-L/14`). The final score is the arithmetic mean over both models.

- **DINO-Score.** Cosine similarity between the CLS tokens of the generated image and the reference image, averaged across four DINOv2 Oquab et al. (2024) variants (`DINOv2-ViT-S/14`, `DINOv2-ViT-B/14`, `DINOv2-ViT-L/14`, `DINOv2-ViT-G/14` ).

- **Diversity.** For each prompt we sample $k$ SVGs, extract DINOv2 Oquab et al. (2024) features (all four models), compute the pairwise cosine similarities, and report $1 - \text{mean(similarity)}$ as a diversity score, averaged across the four encoders.

- **VQA-Score.** Following Hu et al. (2023), we generate a set of question–answer pairs about the content of each image, then ask a vision–language model (VLM) to answer based on the generated raster. The score is the fraction of correct answers, averaged over all prompts.

- **Human Preference Score (HPSv2).** Human Preference Scores Wu et al. (2023b;c); Xu et al. (2023) are widely used to evaluate text-to-image models. These scores predict how likely humans would prefer an image based on large-scale ranking data. We use HPSv2 Wu et al. (2023b) to measure the perceptual quality of our rendered SVG images.

## C  More Results on SGP-CompBench

### C.1  Model Performance on CompBench Throughout Training

As reinforcement learning training progresses, our model demonstrates significant improvement trends on the CompBench benchmark. Table 7 illustrates the performance changes across different training checkpoints (from step 30 to step 780).

### C.2  Full Results on SGP-CompBench

We show the full results on SPG-CompBench in Table 8.

## D  Training Details and Experimental Settings

### D.1  Experimental Setup

All experiments run on a single node with eight NVIDIA H100 GPUs (80 GB each) in BF16 mixed precision; optimisation uses AdamW with a constant learning rate of $1 \times 10^{-6}$, a global batch of 128 captions (8 GPUs $\times$ 16 micro-batches), and gradient-norm clipping at 1.0. We run the RL algorithms without reference model. The implementation builds on the open-source `oat-zero`[1] framework.

For inference, throughout training and evaluation we set temperature 1.0 and top_p 1.0. For inference of baseline models, we set temperature 0.7, keep the other parameters as default, with an exception of Qwen2.5-7B setting temperature 1.0 and top_p 1.0. We run all our experiments on a Nvidia 8xH100 GPU node.

### D.2  Preventing Entropy Collapse

Early experiments with a symmetric PPO clip range of `clip_range` $= 0.2$ drove the policy's token-level entropy to near zero, producing degenerate, highly repetitive SVGs. Following Yu et al. (2025), we adopt *asymmetric* clipping:

$$\texttt{clip\_high} = 0.28, \quad \texttt{clip\_low} = 0.20.$$

---

[1] https://github.com/sail-sg/oat-zero

| Step | Attribute Binding ↑ | | | | Relation ↑ | | | | Numeracy ↑ | | | | Avg ↑ |
|---|---|---|---|---|---|---|---|---|---|---|---|---|---|
| | Color | Shape | Texture | BindAvg | 2D | 3D | Implicit | RelAvg | Total | Item | CPI | NumAvg | |
| 030 | 8.3 | 0.0 | 0.7 | 3.0 | 9.8 | 13.1 | 10.2 | 11.0 | 14.1 | 1.9 | 8.9 | 8.6 | 7.4 |
| 060 | 0.0 | 9.9 | 0.0 | 3.3 | 0.0 | 0.0 | 0.0 | 0.0 | 4.9 | 2.1 | 6.4 | 5.2 | 2.5 |
| 090 | 57.4 | 50.6 | 32.9 | 47.0 | 26.9 | 31.4 | 17.1 | 25.2 | 29.4 | 12.2 | 28.3 | 25.3 | 33.4 |
| 120 | 62.4 | 50.6 | 21.4 | 44.8 | 40.3 | 41.9 | 23.4 | 35.2 | 30.1 | 18.6 | 34.4 | 30.4 | 37.6 |
| 150 | 63.5 | 51.6 | 23.9 | 46.3 | 32.6 | 39.8 | 30.4 | 34.3 | 42.4 | 22.6 | 39.9 | 37.0 | 39.5 |
| 180 | 61.2 | 50.1 | 21.1 | 44.1 | 28.8 | 37.1 | 62.6 | 42.9 | 31.1 | 23.1 | 41.3 | 35.6 | 41.5 |
| 210 | 73.3 | 62.2 | 28.0 | 54.2 | 42.2 | 44.4 | 52.7 | 46.4 | 43.4 | 24.9 | 42.3 | 39.0 | 47.6 |
| 240 | 73.7 | 47.5 | 58.5 | 59.7 | 44.5 | 46.3 | 40.6 | 43.8 | 57.3 | 39.5 | 56.2 | 49.7 | 50.7 |
| 270 | 76.5 | 63.2 | 42.5 | 62.1 | 54.1 | 46.3 | 62.1 | 54.2 | 56.8 | 39.2 | 58.5 | 51.8 | 56.7 |
| 300 | 74.2 | 65.5 | 24.0 | 54.6 | 39.9 | 43.0 | 57.0 | 47.0 | 54.8 | 41.4 | 59.8 | 55.1 | 51.9 |
| 330 | 81.0 | 68.5 | 28.0 | 59.1 | 44.1 | 42.5 | 60.3 | 49.0 | 56.8 | 45.5 | 63.1 | 58.3 | 55.1 |
| 360 | 81.0 | 63.9 | 47.0 | 63.9 | 45.2 | 46.5 | 65.0 | 52.2 | 61.0 | 46.5 | 64.2 | 60.0 | 58.6 |
| 390 | 82.3 | 66.5 | 55.0 | 68.0 | 41.3 | 45.4 | 50.9 | 45.8 | 61.1 | 44.1 | 59.9 | 57.0 | 56.9 |
| 420 | 82.1 | 68.3 | 46.5 | 65.6 | 49.2 | 47.7 | 52.0 | 49.6 | 62.7 | 47.5 | 62.6 | 59.6 | 58.1 |
| 450 | 82.5 | 67.3 | 55.6 | 68.5 | 46.1 | 49.1 | 63.8 | 53.0 | 60.3 | 45.7 | 61.0 | 57.8 | 60.0 |
| 480 | 80.8 | 70.1 | 54.4 | 68.4 | 44.0 | 52.4 | 63.2 | 53.2 | 58.4 | 48.3 | 63.0 | 59.1 | 60.4 |
| 510 | 80.4 | 71.2 | 44.1 | 65.2 | 47.0 | 54.1 | 53.3 | 51.5 | 56.6 | 47.2 | 61.0 | 57.4 | 58.1 |
| 540 | 80.6 | 66.3 | 45.9 | 64.3 | 53.1 | 53.4 | 56.6 | 54.4 | 58.6 | 48.6 | 59.0 | 56.8 | 58.8 |
| 570 | 83.4 | 64.5 | 33.0 | 60.3 | 47.6 | 53.2 | 46.7 | 49.2 | 59.6 | 46.8 | 57.6 | 55.9 | 55.0 |
| 600 | 80.7 | 65.7 | 39.0 | 61.8 | 49.2 | 56.3 | 63.5 | 56.3 | 55.9 | 47.9 | 54.4 | 53.4 | 57.6 |
| 630 | 85.0 | 71.8 | 57.4 | 71.4 | 51.4 | 52.0 | 58.0 | 53.8 | 56.9 | 45.1 | 53.5 | 52.5 | 60.1 |
| 660 | 84.9 | 69.5 | 41.6 | 65.3 | 47.0 | 52.2 | 42.7 | 47.3 | 57.0 | 48.9 | 56.5 | 55.1 | 56.0 |
| 690 | 82.5 | 69.5 | 35.0 | 62.4 | 54.2 | 45.3 | 56.3 | 52.0 | 55.1 | 44.8 | 54.5 | 52.7 | 56.0 |
| 720 | 84.3 | 69.7 | 39.6 | 64.5 | 44.5 | 52.2 | 52.1 | 49.6 | 56.0 | 46.7 | 57.1 | 54.8 | 56.5 |
| 750 | 85.5 | 70.3 | 54.9 | 70.2 | 53.7 | 56.3 | 59.8 | 56.6 | 55.4 | 46.7 | 57.6 | 55.0 | 61.3 |
| 780 | 84.7 | 70.9 | 35.7 | 63.8 | 45.1 | 52.9 | 64.3 | 54.1 | 55.5 | 44.4 | 52.4 | 51.4 | 57.1 |

Table 7: Compositional generation results on SGP–CompBench during training, broken down into attribute binding (color, shape, and texture), relation (2D, 3D, and implicit), and numeracy (total count, item existence, and CPI). Average scores are provided for each category and overall.

The wider positive bound allows larger updates when the advantage $A > 0$, fostering exploration of new token sequences, while the tighter negative bound still prevents destructive policy shifts for $A < 0$. This simple change restores a healthy entropy trajectory without sacrificing stability or requiring additional entropy bonuses.

### D.3 Prohibition of Text Elements

Because CLIP rewards can be gamed by rendering the caption verbatim, we extend the *Format–Validity* reward (Section 4.2.1) with a strict ban on SVG text-rendering tags. Concretely, the parser rejects any output containing `<text>`, `<tspan>` and `<textPath>`. Violation sets $r_{\mathrm{fmt}}(s) = 0$, nullifying downstream perceptual rewards and hence providing a strong learning signal against this exploit. This prohibition is applied consistently during training and evaluation to ensure fair assessment of the model's genuine drawing ability.

## E Additional Experiments and Analysis

### E.1 Comparison between GRPO and PPO

We compare GRPO, a critic-free variant of PPO, against standard PPO on identical settings, with same backbone, reward stack, and data split, to isolate the effect of algorithm choice. For both setting we train for

| Model | Attribute Binding ↑ | | | | Relation ↑ | | | | Numeracy ↑ | | | | Avg ↑ |
|---|---|---|---|---|---|---|---|---|---|---|---|---|---|
| | Color | Shape | Texture | Avg. | 2D | 3D | Implicit | Avg. | Total | Item | CPI | Avg. | |
| *Frontier open-source LLMs* | | | | | | | | | | | | | |
| QwQ-32B | 54.3 | 51.0 | 31.4 | 45.6 | 43.6 | 33.5 | 46.0 | 41.0 | 79.9 | 21.1 | 51.4 | 50.9 | 45.2 |
| DeepSeek-R1 | 72.6 | 62.7 | 48.4 | 61.2 | 59.3 | 43.8 | 58.2 | 53.7 | 83.5 | 35.4 | 60.4 | 57.4 | 57.4 |
| *Frontier closed-source LLMs* | | | | | | | | | | | | | |
| GPT-4o-mini | 60.8 | 52.1 | 39.0 | 50.6 | 39.1 | 36.1 | 42.9 | 39.4 | 80.3 | 19.5 | 37.4 | 45.7 | 44.3 |
| GPT-4o | 62.2 | 48.7 | 34.3 | 48.4 | 49.7 | 37.3 | 49.2 | 45.4 | 85.9 | 25.5 | 51.1 | 52.7 | 48.3 |
| o1-mini | 60.5 | 47.5 | 46.2 | 51.4 | 43.8 | 30.7 | 46.3 | 40.3 | 89.3 | 28.1 | 58.6 | 58.2 | 48.9 |
| o1 | 70.8 | 25.2 | 53.0 | 49.6 | 54.6 | 39.4 | 46.4 | 46.8 | 66.4 | 20.1 | 41.7 | 42.0 | 46.7 |
| o3-mini | 60.5 | 46.7 | 55.1 | 54.1 | 64.7 | 43.9 | 61.6 | 56.8 | 90.8 | 34.5 | 66.4 | 64.7 | 57.7 |
| o3 | 88.9 | 73.6 | 71.7 | 78.1 | 81.6 | 62.0 | 84.5 | 76.0 | 91.6 | 59.8 | 81.1 | 78.8 | 77.5 |
| o4-mini | 82.4 | 62.1 | 69.6 | 71.4 | 71.0 | 57.9 | 76.5 | 68.5 | 90.3 | 52.9 | 76.1 | 74.3 | 71.0 |
| Gemini 2.0 Flash | 58.7 | 49.5 | 37.7 | 48.6 | 43.7 | 31.8 | 40.6 | 38.7 | 85.9 | 24.6 | 52.1 | 54.2 | 47.1 |
| Gemini 2.5 Flash Preview | 63.6 | 45.0 | 56.9 | 55.2 | 46.0 | 38.9 | 57.1 | 47.3 | 82.8 | 34.5 | 62.0 | 59.8 | 53.4 |
| Gemini 2.5 Pro Preview | 88.1 | 65.7 | 74.9 | 76.2 | 77.4 | 59.1 | 80.0 | 72.2 | 94.7 | 68.0 | 83.8 | 82.3 | 76.2 |
| Claude 3.5 Sonnet | 75.3 | 71.2 | 57.1 | 67.9 | 62.0 | 50.4 | 65.0 | 59.1 | 87.1 | 44.5 | 75.0 | 71.3 | 65.4 |
| Claude 3.7 Sonnet | 89.3 | 82.8 | 77.3 | 83.1 | 75.9 | 59.4 | 73.7 | 69.7 | 91.4 | 65.5 | 85.5 | 82.5 | 77.9 |
| Claude 3.7 Sonnet Thinking | 90.5 | 85.6 | 82.4 | 86.2 | 80.2 | 74.4 | 86.4 | 80.3 | 94.9 | 78.9 | 91.4 | 89.4 | 84.8 |
| *Our open-source baseline and RL-tuned models* | | | | | | | | | | | | | |
| Qwen-2.5-7B | 7.1 | 10.0 | 1.7 | 6.3 | 5.2 | 5.8 | 8.1 | 6.4 | 42.6 | 5.8 | 10.7 | 16.1 | 8.8 |
| Qwen-2.5-7B w/ RL **(750)** | 84.3 | 71.3 | 46.0 | 67.2 | 55.7 | 53.9 | 61.7 | 57.1 | 63.4 | 47.5 | 57.6 | 56.8 | 60.8 |
| Qwen-2.5-7B w/ RL **(900)** | 86.3 | 74.9 | 60.7 | 74.0 | 50.4 | 51.1 | 62.4 | 54.6 | 57.5 | 46.6 | 55.0 | 53.8 | 61.7 |

Table 8: Compositional generation results on SGP–CompBench, broken down into attribute binding (color binding, shape binding and texture binding), relation (2D relation, 3D relation and implicit realtion), and numeracy (total count, item existenece and count per item (CPI)). Average scores are provided for each category and overall.

| Algorithm | CLIP ↑ | | | DINO ↑ | | | VQA ↑ | | | HPS ↑ | | | Diversity ↑ | | |
|---|---|---|---|---|---|---|---|---|---|---|---|---|---|---|---|
| | COCO | SGP | Avg. | COCO | SGP | Avg. | COCO | SGP | Avg. | COCO | SGP | Avg. | COCO | SGP | Avg. |
| GRPO | **0.259** | **0.283** | **0.271** | **0.089** | **0.555** | **0.322** | **0.596** | **0.522** | **0.559** | **0.159** | **0.177** | **0.168** | 0.178 | 0.198 | 0.188 |
| PPO | 0.242 | 0.276 | 0.259 | 0.083 | 0.532 | 0.308 | 0.511 | 0.510 | 0.511 | 0.157 | 0.142 | 0.150 | **0.249** | **0.245** | **0.247** |

Table 9: Comparison of GRPO and PPO after 1,020 steps on COCO-val and SGP-Object-val. GRPO achieves better alignment and semantic accuracy.

1 020 steps on the identical Qwen2.5-3B backbone. The results are presented in Table 9. In general, GRPO outperforms PPO in all metrics, while PPO yields higher diversity.

### E.2 Effect of Training–Data Mixture

To disentangle the influence of training corpora from that of reward models, we **fix the reward stack to *SigLIP Base/16-384* and *omit* the DINOv2 model** and vary only the ratio of COCO captions to MMSVG-Illustration-40k captions:

- **Baseline (50 % COCO / 50 % MMSVG)** – the mix used in all previous experiments;

- **100 % COCO** – pure natural-image captions;

- **100 % MMSVG** – pure synthetic SVG captions.

We evaluate on the validation splits (COCO–Val, SGP–Single-9k-val) and report **CLIP**, **DINO**, **VQA**, and **Diversity**. We train each setting to step 750 for fair comparison.

**Discussion.** We summarize patterns from Table 10. Models trained on a *single* corpus score highest on their own validation set but drop sharply on the opposite set. Training in the **50 / 50 mixture** sacrifices

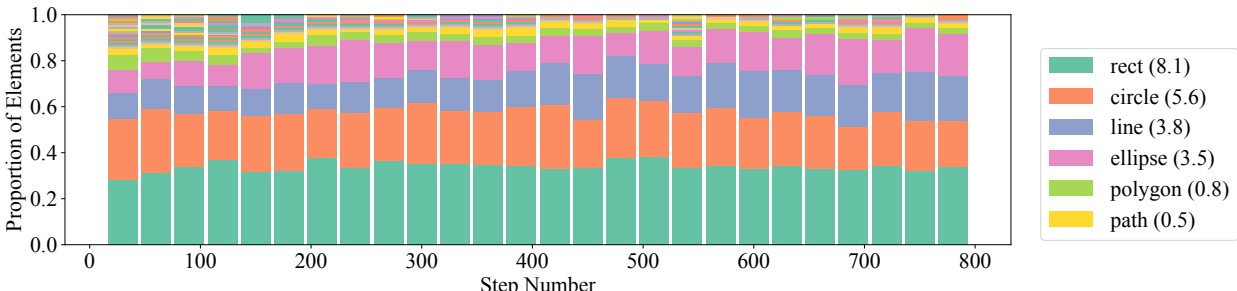

Figure 11: Evolution of SVG element type distribution throughout training. The numbers behind each legend denotes the average numbers of this element per SVG code across steps.

slightly in each domain, yet produces a more balanced overall performance. Since natural-image captions (COCO) emphasize rich scene semantics, whereas SVG captions (MMSVG) emphasize explicit geometry, combining them supplies mutually reinforcing signals that single-domain training lacks. We thus conclude that the gains from the mixed data suggest that expanding caption–style coverage is a promising route to further improvements. It also shows that the cross-domain generalization is limited. In other words, breadth of data is important for SVG-generation tasks.

| Train Mix | | CLIP↑ | | | DINO↑ | | | VQA↑ | | | Diversity↑ | | |
|---|---|---|---|---|---|---|---|---|---|---|---|---|---|
| COCO | MMSVG | COCO | SGP | Avg. | COCO | SGP | Avg. | COCO | SGP | Avg. | COCO | SGP | Avg. |
| 100% | 0% | **0.265** | 0.278 | **0.272** | **0.110** | 0.485 | 0.298 | **0.664** | 0.529 | **0.597** | 0.174 | **0.240** | 0.207 |
| 50% | 50% | 0.258 | 0.286 | **0.272** | 0.102 | 0.566 | **0.334** | 0.632 | 0.560 | 0.596 | 0.184 | 0.194 | 0.189 |
| 0% | 100% | 0.228 | **0.287** | 0.258 | 0.050 | **0.570** | 0.310 | 0.440 | **0.563** | 0.502 | **0.230** | 0.194 | **0.212** |

Table 10: Comparison of training–data mixtures. Bold numbers denote the best value in each column.

### E.3 Element Type Distribution Shifts During Training

Figure 11 reveals the evolution of SVG element usage throughout training. We observe that dominant elements such as `<rect>`, `<circle>`, `<line>`, `<ellipse>`, `<polygon>`, and `<path>` maintain their prevalence, while various less frequently used elements show a gradual decline in usage.

### E.4 Model Behaviors

We found the RL-trained policy exhibits a camera-like framing strategy: it frequently places shapes at absolute coordinates that extend well beyond the declared `viewBox`. Because any path segment outside the `viewBox` is silently clipped by the SVG renderer, the model effectively uses the `viewBox` as a virtual viewfinder, "cropping out" the surplus geometry and revealing only the portion that aligns with the caption (Figure 12). This behavior emerges without any explicit instruction—suggesting the optimizer discovered that overshooting object boundaries is cheaper (in token entropy) than fine-tuning every coordinate to fit perfectly inside the frame. The result is visually coherent images with cleaner edges and fewer near-miss truncation artefacts.

### E.5 Evolution of Chain-of-Thought

Unlike token-level supervised fine-tuning, reinforcement learning encourages the model to verbalize *why* its drawing matches the caption. As shown in Table 11, early in training the policy already learns to reason about how to decompose the scene into different objects and further decompose the objects into components. Later on the optimization steps nudge it toward richer details: detailed description of some key components, relations between objects, activities of persons, and even lighting and shading. These CoTs act as an internal checklist that aligns the SVG with the caption.

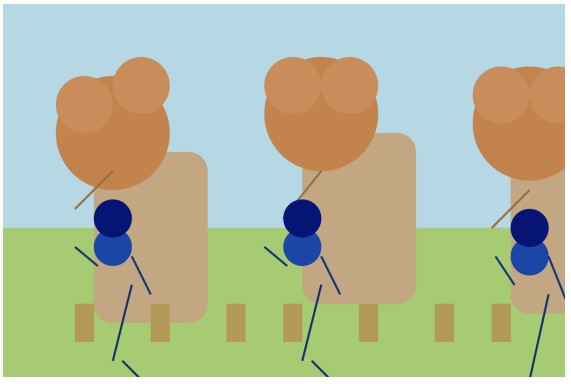 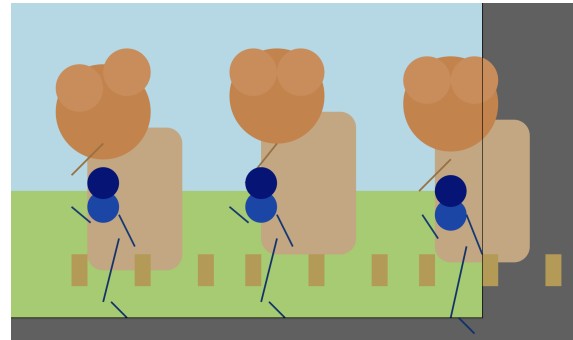

Figure 12: Qualitative examples from the model that draws a full elephant whose body extends outside the `viewBox` (right). The renderer clips the excess, yielding a neatly framed composition (left).

| Step | Chain-of-Thought | Rendered image |
|:---:|:---:|:---:|
| 30 | To create an SVG image of a person driving a plated motorcycle on a track with people watching, we need to include several elements such as a track, a motorcycle, a person, and some onlookers. The motorcycle will have wheels, a body, and plates. The track can be represented as a curve or straight lines depending on the desired style. People can be drawn as simple stick figures or more detailed figures with bodies, heads, and arms. We will use various SVG shapes like circles for wheels, rectangles for bodies, and lines for limbs. | |
| 60 | To create an SVG image of a person driving a plated motorcycle on a track with people watching, we need to consider individual components such as the track, the ground beneath it, the motorcycle with its features (like wheels, body, seat, handlebars, and headlight), the driver inside the motorcycle, spectators watching the race. Different shapes and colors will be used to depict each component accurately. | |
| 120 | In order to create an SVG image for a person driving a plated motorcycle on a track with people watching, we need to consider the shapes and components involved for the motorcycle, the body, wheels, seat, rider, track, and crowd. For the track, we need to consider the boundary and the grooves. The crowd needs appropriate head shapes, bodies, and legs. | |
| 300 | To generate an image of a person driving a plated motorcycle on a track with people watching with the use of SVG, I should consider defining shapes, colors, and proportions for each element. The track will have curves and banking to accurately portray the course. The motorcycle should have features like wheels, handlebars, frame, body, and tires. The person should consist of a head, torso, and limbs. The spectators should each have distinct shapes for heads, bodies, and faces. The lighting and shading should be taken into account for more realistic representation. | |
| 750 | To generate an image showing a person driving a plated motorcycle on a track with spectators watching, I should incorporate geometric forms depicting a road and track, motorcycle, human figures, spectators. I must incorporate some recognizable components such as frame, seat, wheels, body, handlebars, rider, and motorcycle. I should reproduce or construct recognizable sightseeing components on or around a racetrack. I should include features that illustrate or emulate spectators watching. I should create recognizable, clutchable geometries and recognizable features. | |

Table 11: Evolution of our RL-trained model's Chain-of-Thought on the caption *"Person driving a plated motorcycle on a track with people watching."*

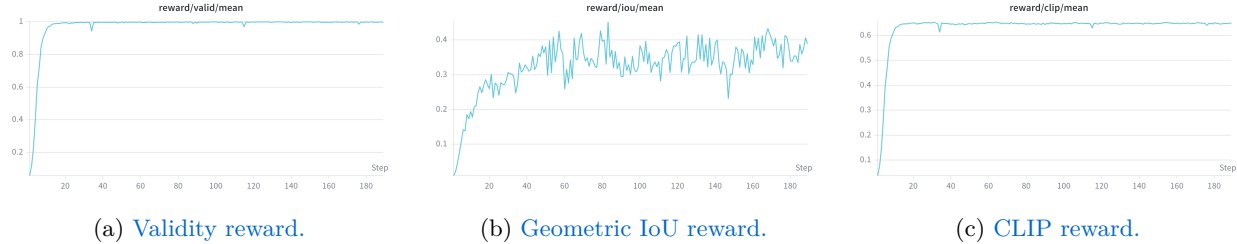

(a) Validity reward.      (b) Geometric IoU reward.      (c) CLIP reward.

Figure 13: Training curves for CAD generation. From left to right: validity reward, geometric IoU reward, and CLIP reward. The validity reward converges rapidly, while the geometric and semantic rewards both improve steadily during RL training.

### E.6    RLVR on CAD Generation

To evaluate whether our RL training framework generalizes beyond SVG generation, we conducted experiments on the domain of 3D CAD code generation. In this setting, the model is trained to produce executable `CadQuery` Python programs that reconstruct a target 3D shape from its natural language description.

We trained the model with two rewards. First, we used a geometric IoU reward, which measures the geometric overlap between the size-normalized generated shape and the size-normalized ground-truth shape. Second, we used a CLIP-based reward, which measures the alignment between the rendered image of the generated shape and the caption text. In addition, we applied a format gate so that only generations that can be successfully executed as valid Python `CadQuery` code receive reward.

For the training data, we combined three datasets: DEEPCAD, which provides CAD assets in CAD command form; TEXT2CAD, which annotates these shapes with natural language descriptions at different levels of abstraction; and TEXT2CADQUERY, which translates CAD commands into Python programs using the `cadquery` package. We selected the abstract caption level and compiled a dataset of paired `CadQuery` Python programs and text descriptions of 3D shapes. The resulting dataset contains approximately 70k samples. We use Qwen-2.5-3B as the base model in this experiment.

Figure 13 shows the training curves of the three reward components. We observe that the format-gate reward converges very quickly, rising from approximately 0 to more than 99% within the first 20 steps. At the same time, the geometric IoU reward increases from approximately 0 to around 0.35, while the CLIP reward improves from approximately 0 to around 0.65. These trends indicate that the proposed RLVR training framework remains effective beyond SVG generation and can be successfully applied to the CAD generation domain.

We further present qualitative examples in Table 12. The generated shapes are generally consistent with the target shapes under rendering, showing that the model can learn to produce executable CAD programs that capture both coarse geometry and textual semantics. For example, the model is able to generate simple primitives such as cylindrical objects and box-like shapes with reasonable overall structure. These results further support the applicability of our method to CAD generation.

### E.7    Formal Correctness Validation

To address the concern of evaluation circularity (i.e., reward and evaluation relying on similar VLM signals), we introduce an independent, executable formal checker that does not use reward-model logits or VLM

| Prompt | Generated v.s. ground truth |
|---|---|
| Use CadQuery to generate executable Python CAD code. Assign the final shape to a variable named result. Do not export or write any files. Output only Python code. Here is the object description: A cylinder with a flat circular top and bottom, and a curved surface connecting them. |  |
| Use CadQuery to generate executable Python CAD code. Assign the final shape to a variable named result. Do not export or write any files. Output only Python code. Here is the object description: A cylindrical object with a smooth surface and rounded edges. |  |
| Use CadQuery to generate executable Python CAD code. Assign the final shape to a variable named result. Do not export or write any files. Output only Python code. Here is the object description: A cube with a circular hole on one of its faces. The cube has a flat top and bottom, and the circular hole is centered on one of the faces. |  |

Table 12: Qualitative CAD generation results. For each example, we show the input prompt and a comparison between the ground-truth and generated renderings.

similarity scores at evaluation time. Our benchmark is relation-centric: every sample defines two objects and one explicit spatial relation, and the generated SVG must satisfy deterministic program-level constraints after parsing. Concretely, we evaluate six relation categories: `left_of`, `right_of`, `above`, `below`, `overlap`, and `non_overlap`. These six categories cover both directional ordering (`left_of`/`right_of`/`above`/`below`) and geometric interaction (`overlap`/`non_overlap`), which together capture the core instruction-following behavior for compositional 2D layout generation.

Shape and color constraints are used primarily as object-binding anchors for formal verification. In other words, we do not add shape/color complexity for visual diversity alone; we require distinguishable attributes so that the checker can unambiguously bind language entities (e.g., "the purple square" vs "the blue circle") to concrete SVG elements before relation verification. This design directly prevents degenerate relation scoring where the system cannot reliably decide which rendered element corresponds to which textual object. After binding, relation correctness is evaluated geometrically (center-based direction checks and overlap/non-overlap checks), yielding reproducible pass/fail outcomes with explicit error traces.

Empirically, this formal protocol provides stronger and more interpretable evidence than similarity-only metrics. The closed-source frontier models are near-saturated (e.g., *claude-sonnet-4-20250514* at 99.50% overall and *gpt-4o* at 98.50%), while smaller/open models show clear category-specific weaknesses, especially on directional reasoning and non-overlap edge cases. Importantly, our RL-trained model (trained from a Qwen2.5-7B base with CLIP reward) substantially improves over its baseline family in both overall formal pass rate (41.33% to 72.17%) and multiple fine-grained categories, supporting the claim that reward-based training improves not only perceptual alignment but also formal instruction-following and generalization under symbolic checks.

| Model | Parse | Above | Below | Left-of | Non-overlap | Overlap | Right-of | Overall. |
|---|---|---|---|---|---|---|---|---|
| *Frontier Closed-source LLMs* | | | | | | | | |
| Claude-4-Sonnet | 100.00 | 98.00 | 99.00 | 100.00 | 100.00 | 100.00 | 100.00 | 99.50 |
| Gpt-4o | 100.00 | 100.00 | 97.00 | 100.00 | 94.00 | 100.00 | 100.00 | 98.50 |
| Gpt-5.4 | 100.00 | 78.00 | 75.00 | 99.00 | 99.00 | 100.00 | 100.00 | 91.83 |
| Gpt-4.1 | 100.00 | 86.00 | 80.00 | 84.00 | 99.00 | 100.00 | 92.00 | 90.17 |
| Gpt-5.2 | 100.00 | 75.00 | 84.00 | 72.00 | 100.00 | 100.00 | 91.00 | 87.00 |
| *Frontier Open-source LLMs* | | | | | | | | |
| Deepseek-V3 | 99.67 | 62.00 | 93.00 | 81.00 | 85.00 | 100.00 | 82.00 | 83.83 |
| Qwen-2.5-7B w/RL | 100.00 | 55.00 | 85.00 | 100.00 | 14.00 | 79.00 | 100.00 | 72.17 |
| Qwen-2.5-7B-Inst. | 92.33 | 41.00 | 52.00 | 69.00 | 3.00 | 73.00 | 87.00 | 54.17 |
| Qwen-2.5-7B | 100.00 | 9.00 | 23.00 | 63.00 | 0.00 | 90.00 | 63.00 | 41.33 |

Table 13: Evaluation results of different models on the formal relation-centric benchmark. The RL-tuned model demonstrates significant improvements over its base model.

