# OpenReview forum: "Symbolic Graphics Programming with Large Language Models"
_TMLR — Decision pending for TMLR_

### Review · Reviewer_FsV7 · 2026-01-22

**Summary Of Contributions:**

This paper proposes a new benchmark SGP-GenBench, which aims to measure the capability of frontier LLMs to generate scalable vector graphics (SVG) images. This is an indicator of both the LLM's visual understanding and its code generation ability. In addition to the benchmark, the paper contributes a thorough evaluation of frontier LLM performance on the tasks, and also proposes a GRPO-style reinforcement learning approach to bring open-source LLMs up to frontier-level performance on SVG generation. The paper also contributes an analysis of RL dynamics, revealing emergent behaviors like finer object decomposition and added contextual details for scene coherence.

Key strengths:
- Provides an interpretable way to probe LLMs' visual grounding through symbolic programs, with strong emphasis on precision and controllability over pixel-based methods.
- RL approach is scalable and leverages existing vision encoders without new parameters, closing the gap for open-source models.
- Qualitative and statistical analyses offer insights into model evolution during training.

Key weaknesses:
- Focuses exclusively on SVGs and the generated images are fairly simplistic.
- Proposed model does not have any vision encoder, so the model cannot check and iterate upon its own samples.
- Somewhat unclear novelty / contribution compared to many related works such as StarVector and follow ups.

**Audience:**

Yes

**Audience Explanation:**

The findings advance our understanding of LLMs' cross-modal capabilities, particularly in bridging language to visual synthesis via symbolic code. This is relevant to TMLR readers interested in program synthesis, multimodal grounding, and RL finetuning for visual tasks. The benchmark and RL method could inspire applications in graphics generation, interpretable AI, and open-source model enhancement, especially as SVGs could potentially offer advantages in precision and controllability over diffusion-based image gen.

However, the paper's positioning could benefit from clarification given the many related works doing similar things, including multimodal LLMs generating SVGs with reinforcement learning.

**Broader Impact Concerns:**

One major limitation of the proposed method is that there is no mechanism for the model to visually check its own output, as a human would do. The model is forced to generate the SVG in one single shot. A more interesting and realistic setting might be to do the rendering in multiple passes, using a VLM instead of LLM in order to visually encode its own draft renderings and improve them iteratively.

**Claims And Evidence:**

Yes

**Claims Explanation:**

The paper presents comprehensive evaluations comparing frontier LLMs as well as open source LLMs on the new benchmark. The model training and SVG generation process is explained clearly and builds on well established methods. Qualitative results indicate an evolution of model strategies over the course of RL post-training, showing gradual improvement of visual fidelity and conceptual structure in the generated SVG.
Additionally, quantitative results in Tables 1 and 2 (and full results in Section 5.2) use multiple metrics to back claims of proprietary model superiority and RL improvements (e.g., Qwen-2.5-7B's compositional score jumps from 8.8 to 60.8). The RL pipeline is detailed in Figure 4, with ablation studies in the appendix supporting reward component choices. Case studies in Figure 1 and statistical analyses (e.g., program length, primitive decomposition) provide clear evidence for emergent behaviors. Overall, the evidence looks reproducible, with benchmarks and methods described in sufficient detail.

**Requested Changes:**

- Add more discussion about the comparison to SVG-Bench (https://arxiv.org/pdf/2312.11556) and StarVector. How does SVG-GenBench differ specifically? It looks as though StarVector also does text2SVG, in addition to image2SVG, although they do not do reinforcement learning.
- "Rendering-Aware Reinforcement Learning for Vector Graphics Generation" also uses GRPO-style reinforcement learning for text-to-SVG generation. What are the main contributions of SVG-GenBench going beyond this paper?

---

> ### Author Response · Authors · 2026-03-14
>
> We sincerely thank the reviewer for the constructive comments on our work. We take every comment seriously and hope our response can address the reviewer’s concerns. If there are any remaining questions, we are happy to address them. We summarize all the concerns into the following questions:
>
> **Q1: Comparison to SVG-Bench and StarVector**
>
> **A1:** We thank the reviewer for pointing us to StarVector and SVG-Bench. We will clarify the comparison more carefully in the revision. StarVector introduces a multimodal SVG generation model together with SVG-Bench, which evaluates three task categories: image-to-SVG, text-to-SVG, and diagram generation. Our work differs in both training setting and evaluation goals. On the training side, our RL framework improves an LLM for symbolic graphics programming using caption supervision plus rendered-image rewards, without requiring paired ground-truth SVG programs. On the evaluation side, SGP-GenBench is organized by capability axes of object fidelity, scene fidelity, and compositional consistency, because our goal is to analyze which aspects of symbolic graphics programming current LLMs can or cannot perform. We will make this distinction clearer and cite StarVector/SVG-Bench explicitly in the related-work comparison.
>
> **Q2: Focuses exclusively on SVGs and the generated images are fairly simplistic; proposed model does not have any vision encoder, so the model cannot check and iterate upon its own samples.**
>
> **A2:** We agree that our empirical scope is currently limited to SVG, and that SVG outputs are structurally simpler than full natural-image generation. This restriction is intentional: SVG provides deterministic rendering, precise parametric control, and direct program interpretability, making it a clean testbed for studying symbolic graphics programming. Importantly, our benchmark goes beyond simple icons by covering object-level, scene-level, and compositional generation, and our analysis shows that RL leads to longer programs, finer decomposition, and richer contextual details over training.
>
> We also agree that our model does not contain an internal vision encoder and therefore cannot explicitly render, inspect, and revise its own outputs in a closed loop. In this work, that is a deliberate design choice rather than an oversight: we ask whether a pure code-generating LLM can acquire visual grounding through fixed external reward models, without changing the architecture. With this restriction, we are seamlessly exploring the possibility of encoding the visual signals as some structured language (either natural or formal), which is a natural way for a unified understanding of visual and textual inputs. With that being said, we still view integrating an internal vision module for iterative self-correction as an important and complementary future direction.

---

> ### Author Response · Authors · 2026-03-14
>
> **Q3: "Rendering-Aware Reinforcement Learning for Vector Graphics Generation" also uses GRPO-style reinforcement learning for text-to-SVG generation. What are the main contributions of SVG-GenBench going beyond this paper?**
>
> **A3:** Our work is concurrent with "Rendering-Aware Reinforcement Learning," as both were submitted to NeurIPS 2025. While both approaches utilize similarity scores from vision encoders as rewards for RLVR training, our work, SVG-GenBench, offers several distinct contributions that go beyond their scope:
>
> **Evaluation of Compositional Reasoning (SGP-CompBench):** Rather than focusing solely on image quality, we treat SVG as a procedural, compositional Domain-Specific Language (DSL). We introduce SGP-CompBench (Section 5.2.2) to evaluate the model’s ability to handle attribute binding, spatial relationships, and numeracy—dimensions of structural reasoning not explored by concurrent works.
>
>
> **Analysis of Continuous Reward Scaling (Section 6.1):** We provide a rigorous analysis of how continuous CLIP rewards differ from the binary "correct/incorrect" rewards used in mathematical RLVR. We demonstrate that while base models in math often catch up to RL policies at n≈$2^9$, our RL policy maintains its superiority until n exceeds $10^6$. This massive crossover point proves that our approach yields a fundamental leap in reasoning rather than a simple narrowing of the output distribution (policy sharpening).
>
>
> **Discovery of Emergent Strategies (Section 6.2):** We identify two sophisticated strategies that emerge during training to improve SVG quality:
> - **Strategy 1 (Decomposition):** The model learns to decompose complex objects into basic primitives, achieving superior geometric precision through better spatial composition rather than merely expanding its vocabulary of shapes.
> - **Strategy 2 (Contextual Details):** Beyond literal prompt fulfillment, the policy spontaneously introduces semantically consistent details (e.g., adding waves to a beach scene) that enhance the overall coherence and realism of the generated graphic.

---

### Review · Reviewer_dzcG · 2026-02-06

**Summary Of Contributions:**

The paper proposes (1) a benchmark for SVG-generation capabilities of LLMs; and (2) an RL optimization framework for promoting such capabilities in LLMs without using any paired {prompt, golden-SVG} data. The paper benchmarks various open-source and closed-source LLMs on their benchmark and further conduct detailed experiments and ablations regarding their proposed optimization framework.

Positives:
- Presents a new benchmark for SVG-creation capabilities of LLMs
- Proposed CLIP-based reward function doesn’t require paired {prompt, SVG / reference image} data
- Paper is well-written and easy to follow

Negatives:
- A relatively niche problem that doesn’t have a large audience
- The novelty of the paper is limited to the domain-specific reward formulation & corresponding empirical study
- The paper doesn’t compare / mention with a very similar, recently published approach at NeurIPS ‘25 [1] raising even further novelty concerns

Questions / comments:
- The paper mentions the reward formulation as a verifiable reward. While SVG compilation might be called as a verifiable reward source, SigLIP- / DINO-based similarity rewards don’t seem to fit in into the verifiable reward formulation
- How does the proposed technique based on SigLIP / DINO-based rewards compare to the LLM-as-a-judge based reward setup for the SVG generation task proposed in [1]?

[1] Rodriguez, Juan A., et al. "Rendering-Aware Reinforcement Learning for Vector Graphics Generation." NeurIPS 2025. https://openreview.net/forum?id=2Twzlf6qFv

**Audience:**

No

**Audience Explanation:**

Arguable here - while the paper is already intended for a niche audience, my primary reason to vote for “No” here is the existence of an already published method that offers very similar insights to this paper.

**Claims And Evidence:**

Yes

**Claims Explanation:**

Yes, the paper’s proposed hypothesis is well supported and justified by various experiments and ablations.

**Requested Changes:**

An in-depth comparison of the proposed technique with the relevant NeurIPS ‘25 paper. Please also see the minor questions above.

---

> ### Author Response · Authors · 2026-03-14
>
> We sincerely thank the reviewer for the constructive comments on our work. We take every comment seriously and hope our response can address the reviewer’s concerns. If there are any remaining questions, we are happy to address them. We summarize all the concerns into the following questions:
>
> **Q1: The paper mentions the reward formulation as a verifiable reward. While SVG compilation might be called as a verifiable reward source, SigLIP- / DINO-based similarity rewards don’t seem to fit in into the verifiable reward formulation.**
>
> **A1:** We thank the reviewer for this important clarification. We agree that the term “verifiable reward” can be interpreted too broadly in the current draft. In our method, the strictly verifiable component is the binary format-validity gate: the output must satisfy the required structure and compile/render successfully as SVG. By contrast, the SigLIP- and DINO-based terms are automated perceptual similarity rewards from fixed pretrained encoders, not formal verifiers in the hard rule-based sense.
> Our reward is therefore better characterized as a hybrid reward formulation: a deterministic verifiable gate multiplied by foundation-model-based perceptual rewards. We will revise the wording throughout the paper to avoid overstating the verifier aspect and to distinguish clearly between (i) renderability/format verification and (ii) semantic alignment rewards from external vision encoders.
> Concretely, we will replace phrases such as “RL with verifiable rewards” with more precise wording such as “RL with a verifiable format-validity gate and cross-modal perceptual rewards” or “hybrid verifier-plus-perceptual reward training.” This change better reflects the method and addresses the reviewer’s concern.
>
>
>
> **Q2: How does the proposed technique based on SigLIP / DINO-based rewards compare to the LLM-as-a-judge based reward setup for the SVG generation task ?**
>
> **A2:** We thank the reviewer for this comparison. Our paper does not include a direct head-to-head training comparison against an LLM-as-a-judge reward, so we do not claim empirical superiority over that setup. Instead, we view the two reward families as complementary. The proposed SigLIP/DINO-based reward provides a dense perceptual signal on rendered SVGs: SigLIP supplies text-image alignment for all prompts, while DINO can additionally measure image-image fidelity when a reference image is available. This makes the reward lightweight, general, and applicable to open-domain SVG generation. In contrast, an LLM-as-a-judge reward is likely better suited to discrete constraints such as exact object counts or spatial relations, where global embedding similarity is less precise. This is also consistent with our evaluation design: on the compositional split, we use judge-based evaluation specifically because attribute binding, relations, and numeracy require more explicit checking than CLIP/VQA-style metrics alone. We will clarify in the revision that our method primarily targets broad semantic grounding and visual fidelity, while judge-based rewards are a promising complementary direction for stricter compositional supervision.
>
> We also clarify that the rendering-aware RL$^{[1]}$ work does not appear to use a VLM as the training reward model for Text2SVG. As stated in that paper, CLIP is used as the reward to optimize the similarity between the text prompt and the rendered SVG image, while the VLM is used as a judge to assess generation quality. Therefore, our comparison is not between a vision-encoder reward and a VLM-based training reward, but rather between two complementary roles: dense perceptual rewards for optimization versus judge-based assessment for stricter checking. This is also consistent with our own setup: our training reward is based on SigLIP/DINO-style perceptual alignment, while judge-based evaluation is used on the compositional split, where exact attribute binding, spatial relations, and numeracy are better captured by explicit judgment than by global embedding similarity alone.
>
> [1] Rodriguez, Zhang, et al. "Rendering-Aware Reinforcement Learning for Vector Graphics Generation"

---

### Review · Reviewer_cXvp · 2026-02-19

**Summary Of Contributions:**

This paper studies symbolic graphics programming (SGP), where large language models (LLMs) generate executable SVG programs from natural language descriptions. Unlike pixel-based image generation, this task requires producing structured programs that precisely encode visual semantics, providing an interpretable testbed for evaluating cross-modal grounding and compositional reasoning.

The authors introduce SGP-GenBench, a benchmark evaluating object fidelity, scene fidelity, and compositionality (attribute binding, spatial relations, and numeracy). They find that closed-source models significantly outperform open-source models and that performance correlates with general coding ability.

To improve open-source performance, the paper proposes a reinforcement learning (RL) approach using verifiable rewards based on format validity and cross-modal alignment between rendered SVG images and text or reference images. Fine-tuning Qwen-2.5-7B with this method yields substantial gains, closing much of the gap with proprietary models.

Overall, the paper presents a benchmark, training method, and empirical analysis showing that reinforcement learning with cross-modal rewards significantly improves LLM capabilities in generating semantically aligned symbolic graphics programs.

**Audience:**

No

**Audience Explanation:**

Symbolic graphics programming provides a structured and interpretable setting for studying cross-modal grounding and program synthesis, which are both central topics in machine learning. By framing visual generation as executable program generation rather than pixel synthesis, the paper offers a precise testbed for evaluating compositional reasoning, semantic alignment, and the effects of reinforcement learning with verifiable rewards. The proposed benchmark and empirical findings—particularly the demonstration that reinforcement learning with cross-modal similarity rewards substantially improves open-source models—are likely to be of interest to researchers working on multimodal learning, reinforcement learning post-training, program synthesis, and LLM evaluation. Additionally, the benchmark itself may serve as a useful resource for future work on interpretable and structured visual generation.

**Claims And Evidence:**

Yes

**Claims Explanation:**

Strengths

- **Well-motivated problem and clean evaluation setting.** Symbolic graphics programs provide structured, interpretable, and verifiable outputs, offering a precise testbed for evaluating cross-modal grounding and compositional generation.

- **Useful benchmark contribution.** The proposed SGP-GenBench evaluates object fidelity, scene fidelity, and compositional reasoning, and could serve as a valuable resource for future research.

- **Effective and scalable training approach.** The RL framework leverages verifiable rewards from vision-language models without requiring ground-truth SVG annotations, and shows substantial improvements over the base model.

- **Strong empirical evaluation and analysis.** The paper demonstrates meaningful performance gains and provides qualitative and quantitative insights into how RL improves compositional and semantic generation.

**Requested Changes:**

**Important Changes I would like to see**

*1. Clarify novelty relative to prior RLVR and multimodal RL work.* The RL framework largely follows existing approaches using verifiable rewards from pretrained vision-language models. The authors should more clearly articulate the methodological novelty and distinguish their contribution beyond applying known RL techniques to SVG generation.

*2. Address evaluation circularity and reward-model bias.* The same or similar vision-language models are used for both training rewards and evaluation metrics, raising concerns about overfitting to the reward signal. The authors should include independent evaluation methods, such as human evaluation or formal correctness checks, to demonstrate genuine improvements.

*3. Provide stronger evidence of true compositional and semantic generalization.* It remains unclear whether the model learns meaningful symbolic reasoning or simply optimizes embedding similarity. Additional evaluation on out-of-distribution or compositional generalization tasks would strengthen the claims.

**Adjustments that would strengthen the work**

*4. Provide deeper analysis of training effects.* More systematic analysis of how RL changes program structure, compositional depth, or correctness would improve understanding of the mechanism behind the observed improvements.

*5. Discuss generalization beyond SVG.* The paper should clarify whether the approach extends to broader symbolic program synthesis domains or is specific to SVG generation.

*6. Include more formal correctness-based evaluation.* Since SVG programs are structured and executable, evaluating geometric or structural correctness directly would provide stronger evidence than similarity-based metrics alone.

---

> ### Author Response · Authors · 2026-03-14
>
> **Q1:Clarify novelty relative to prior RLVR and multimodal RL work. The RL framework largely follows existing approaches using verifiable rewards from pretrained vision-language models. The authors should more clearly articulate the methodological novelty and distinguish their contribution beyond applying known RL techniques to SVG generation.**
>
> **A1:** We thank the reviewer for this suggestion. We agree that our method does not introduce a new generic RL optimizer beyond existing RLVR/GRPO-style approaches. Our contribution is instead at the level of problem formulation, reward design in a new structured visual-program domain, benchmark construction, and analysis.
>
> First, we study symbolic graphics programming as a distinct multimodal generation problem, where the model must produce an executable SVG program rather than pixels. This setting is materially different from prior multimodal RL work because the output is a structured, renderable program with strong parametric and procedural constraints, and our paper introduces SGP-GenBench to evaluate this capability across object fidelity, scene fidelity, and compositionality.
>
> Secondly, in contrast to most RLVR works in mathematical settings that rely on binary correctness, our approach utilizes continuous CLIP rewards to avoid the reward sparsity. This helps us to obtain substantive capability gains in SVG generation.
>
> Further, beyond performance gains, we provide evidence that RL is doing more than just a sharpening mechanism. Empirically, RL substantially improves the base Qwen-2.5-7B model on both fidelity and compositionality benchmarks. We evaluate this through a Best-of-n analysis, where the model selects the highest-rewarded response from n candidates. In typical mathematical benchmarks, the base model’s performance (Pass@k) often converges with or even surpasses RL-trained models once k reaches approximately 512 ($2^9$)$^{[1]}$. However, in our SVG synthesis task, our RL checkpoint maintains its superiority until n exceeds 1,000,000 ($10^6$). This massive crossover point demonstrates that matching our RL-tuned policy would require an impractically large number of samples from the base model, confirming a fundamental leap in visual reasoning and structural composition rather than mere policy sharpening.
>
> Finally, our analysis (Strategy 1, Section 6.2) reveals that the model develops an inherent sense of compositionality through the RL process. Specifically, we observe a transition in the policy's generation strategy: whereas early-stage models often produce a single, monolithic block of SVG code for a complex prompt, the fully-trained policy learns to decompose intricate objects into distinct, manageable sub-components. Each component is typically preceded by a natural language comment describing the sub-concept, followed by the corresponding SVG implementation. This hierarchical approach allows the model to handle geometry and spatial positioning for each part individually, resulting in significantly higher global accuracy and more precise spatial relationships. This emergence of 'comment-then-code' modularity serves as strong evidence that the model has progressed beyond simple pattern matching to a compositional understanding of the visual task.
>
> We will revise the paper to make this positioning more explicit and to better distinguish our contributions from prior RLVR and multimodal RL work.
>
> [1] Yue, Yang, et al. "Does reinforcement learning really incentivize reasoning capacity in llms beyond the base model?."

---

> ### Author Response · Authors · 2026-03-14
>
> **Q2: Address evaluation circularity and reward-model bias. The same or similar vision-language models are used for both training rewards and evaluation metrics, raising concerns about overfitting to the reward signal. The authors should include independent evaluation methods, such as human evaluation or formal correctness checks, to demonstrate genuine improvements.**
>
> **A2:** We thank the reviewer for raising this important point. We agree that **evaluation circularity** is a potential concern in reward-based generation. In our setting, the training reward is primarily based on **SigLIP** (text-image alignment) and **DINO** (image-image reward), whereas evaluation utilizes several distinct signals: **CLIP-Score, DINO-Score, VQA-Score, HPSv2**, and a separate compositional benchmark. Thus, the reported gains are not confined to the training reward itself. In particular, the RL-tuned model improves substantially on external metrics such as **VQA** and **LLM-as-judge Score** in SGP-CompBench, which are not directly optimized during training.
>
> To address the concern of evaluation circularity, we introduce an **independent, executable formal checker** that does not use reward-model logits or VLM similarity scores at evaluation time. Our benchmark is **relation-centric**: every sample defines two objects and one explicit spatial relation, and the generated SVG must satisfy deterministic program-level constraints after parsing. Concretely, we evaluate six relation categories: `left_of`, `right_of`, `above`, `below`, `overlap`, and `non_overlap`. These categories cover both directional ordering and geometric interaction, capturing the core instruction-following behavior for compositional 2D layout generation.
>
> Shape and color constraints are used primarily as object-binding anchors for formal verification. We require distinguishable attributes so that the checker can unambiguously bind language entities (e.g., “the purple square” vs. “the blue circle”) to concrete SVG elements. After binding, relation correctness is evaluated geometrically (center-based direction checks and intersection tests), yielding reproducible pass/fail outcomes with explicit error traces.
>
> Empirically, this formal protocol provides stronger and more interpretable evidence than similarity-only metrics. While closed-source frontier models are near-saturated (e.g., Claude-4-Sonnet at 99.50% and GPT-4o at 98.50%), smaller models show clear weaknesses. **Importantly, our RL-trained model (from a Qwen2.5-7B base) substantially improves over its baseline in overall formal pass rate (41.33% to 72.17%), supporting the claim that reward-based training improves formal instruction-following and generalization under symbolic checks.**
>
> | Model | Parse | Above | Below | Left-of | Non-overlap | Overlap | Right-of | Overall. |
> | :--- | :---: | :---: | :---: | :---: | :---: | :---: | :---: | :---: |
> | **Frontier Closed-source LLMs** | | | | | | | | |
> | Claude-4-Sonnet | 100.00 | 98.00 | 99.00 | 100.00 | 100.00 | 100.00 | 100.00 | 99.50 |
> | Gpt-4o | 100.00 | 100.00 | 97.00 | 100.00 | 94.00 | 100.00 | 100.00 | 98.50 |
> | Gpt-5.4 | 100.00 | 78.00 | 75.00 | 99.00 | 99.00 | 100.00 | 100.00 | 91.83 |
> | Gpt-4.1 | 100.00 | 86.00 | 80.00 | 84.00 | 99.00 | 100.00 | 92.00 | 90.17 |
> | Gpt-5.2 | 100.00 | 75.00 | 84.00 | 72.00 | 100.00 | 100.00 | 91.00 | 87.00 |
> | **Frontier Open-source LLMs** | | | | | | | | |
> | Deepseek-V3 | 99.67 | 62.00 | 93.00 | 81.00 | 85.00 | 100.00 | 82.00 | 83.83 |
> | **Qwen-2.5-7B w/RL** | **100.00** | **55.00** | **85.00** | **100.00** | **14.00** | **79.00** | **100.00** | **72.17** |
> | Qwen-2.5-7B-Inst. | 92.33 | 41.00 | 52.00 | 69.00 | 3.00 | 73.00 | 87.00 | 54.17 |
> | Qwen-2.5-7B | 100.00 | 9.00 | 23.00 | 63.00 | 0.00 | 90.00 | 63.00 | 41.33 |

---

> ### Author Response · Authors · 2026-03-14
>
> **Q3: Provide stronger evidence of true compositional and semantic generalization. It remains unclear whether the model learns meaningful symbolic reasoning or simply optimizes embedding similarity. Additional evaluation on out-of-distribution or compositional generalization tasks would strengthen the claims**
>
> **A3:** Thanks for the suggestion! We agree that stronger evidence for true compositional and semantic generalization is important. Our current paper already includes a dedicated compositional benchmark, SGP-CompBench (Section 5.2.2), which contains 3,200 prompts spanning attribute binding, spatial relations, and numeracy over 80 common objects, and was explicitly designed to evaluate compositional generation rather than raw object fidelity. **The prompts within the SGP-CompBench suite were entirely excluded from our training data; consequently, the resulting compositionality evaluation represents a genuine out-of-distribution performance.** On this benchmark, our RL-tuned model improves the overall score from 8.8 to 60.8, with gains observed broadly across color binding, shape binding, spatial relations, and count-per-item numeracy, rather than on a single metric alone.
>
> **In addition, our conclusions are not based solely on the training reward:** We also evaluate with external metrics including VQA, DINO, HPS, and CLIP (see Section 5.2.1), and the RL-tuned model achieves the best overall VQA score among compared systems, suggesting that the improvement is not merely optimization of embedding similarity.
>
> **Finally, our analysis (Strategy 1, Section 6.2) reveals that the model develops an inherent sense of compositionality through the RL process.** Specifically, we observe a transition in the policy's generation strategy: whereas early-stage models often produce a single, monolithic block of SVG code for a complex prompt, the fully-trained policy learns to decompose intricate objects into distinct, manageable sub-components. Each component is typically preceded by a natural language comment describing the sub-concept, followed by the corresponding SVG implementation. This hierarchical approach allows the model to handle geometry and spatial positioning for each part individually, resulting in significantly higher global accuracy and more precise spatial relationships. This emergence of “comment-then-code” modularity serves as strong evidence that the model has progressed beyond simple pattern matching to a compositional understanding of the visual task.

---

> > ### Author Response · Authors · 2026-03-14
> >
> > **Q4: Discuss generalization beyond SVG. The paper should clarify whether the approach extends to broader symbolic program synthesis domains or is specific to SVG generation.**
> >
> > **A4:** We did the RL experiments on CAD generation and checked the applicability of our method on that domain. We trained with 2 rewards: a geometric IoU reward that measures the geometric overlap between the size-normalized ground truth shape and the generated shape, and a clip reward that measures the alignment of the rendered image of the generated shape and the caption text. Meanwhile, we also applied a format gate to make sure the generated code can be executed. For the dataset, we combined 3 datasets: `DeepCAD` for the CAD assets in CAD commands, `Text2CAD` that caption these shapes with natural language descriptions of different levels of abstractions, and `Text2CADQuery` that translates the CAD Commands into python code that imports the package `cadquery.` We choose the abstraction level of the caption to be abstract, and compile a dataset of python `cadquery` codes and the text description of the 3D shapes, the dataset size is ~70k. We use Qwen-2.5-3B for this experiment.
> >
> > We report  the results as follows:
> >
> > 1. **Reward Curves:** During training, the `format gate` reward converged rapidly from near-zero to over 99% within the first 20 steps. Simultaneously, the `IoU reward` increased from 0 to 0.35, and the `CLIP reward` rose from 0 to 0.65. These metrics demonstrate that our RLVR framework maintains strong **generalizability** beyond SVG synthesis, effectively extending to CAD generation. Detailed curves for the Valid Gate, IoU, and CLIP-score are provided in **Appendix E.5 (RLVR on CAD Generation)** of the revised manuscript.
> >
> > 2. **Qualitative Results:** To further illustrate the model's performance, we have included representative qualitative samples of generated CAD models in **Appendix E.5**. These visualizations highlight the structural fidelity achieved by our RL-tuned model.

---

### Author Response · Authors · 2026-03-14
**Overall Reply**

We sincerely thank the reviewer for the constructive comments on our work. We will take all the comments seriously and revise our manuscript accordingly.

We make a few general responses to some of the common questions. We hope they are helpful for reviewers to understand how our work is motivated and why our work is important.


**Q1: Comparison to related works [1][2] and contributions beyond RLVR-style training**

**A1:** We thank the reviewer for the opportunity to clarify our contributions. We will explicitly cite StarVector [1] and the concurrent work on Rendering-Aware RL [2] in our revision. While [2] also applies the RLVR (Reinforcement Learning with Verifiable Rewards) framework to the Text-to-SVG task, our work provides several fundamental contributions that go beyond the standard application of RL with visual rewards:

**1. Comparison with StarVector [1]**

StarVector is primarily a multimodal model trained on supervised, paired SVG datasets. Our work, by contrast, focuses on zero-shot capability enhancement through RL. We do not require ground-truth SVG programs and instead use an LLM's inherent reasoning to bridge the gap between captions and SVG geometry via rendered-image feedback.

**2. Contributions Beyond Standard RLVR-style Training**

While [2] demonstrates the feasibility of RLVR for SVG generation, our study delves deeper into the reasoning mechanics and emergent behaviors that standard RLVR paradigms often overlook:

- **Systematic Evaluation of Compositional Logic (SGP-CompBench):** Unlike [2] which primarily targets visual fidelity, we treat SVG as a procedural Domain-Specific Language (DSL). We introduce SGP-CompBench to isolate and evaluate the model's internal structural reasoning—such as attribute binding, spatial coordination, and numeracy. This allows us to quantify how RL improves the logic of the code, not just the pixels of the output.

- **Rigorous Analysis of Reward Scaling & Reasoning (Section 6.1):** A common critique of RLVR is that it may simply "sharpen" the policy (narrowing the output distribution). We provide a unique analysis showing that while in mathematical RLVR, base models catch up to RL policies at n≈$2^9$ samples, our SVG-RL policy maintains a dominant lead until n>$10^6$. This massive gap is a strong empirical indicator of a fundamental leap in reasoning rather than mere distribution shifting.

- **Discovery of Emergent Visual Strategies (Section 6.2):** We move beyond "prompt-matching" to identify how RLVR triggers high-level cognitive strategies:
    - **Iterative Decomposition:** The model spontaneously learns to break complex visual concepts into basic primitives for geometric precision.
    - **Semantic Contextualization:** The model introduces semantically consistent "bonus" details (e.g., adding waves to a beach) that enhance scene coherence, demonstrating that RLVR can foster creative reasoning that goes beyond literal prompt adherence.

[1] Rodriguez, Juan A., et al. "Starvector: Generating scalable vector graphics code from images and text." Proceedings of the Computer Vision and Pattern Recognition Conference. 2025.

[2] Rodriguez, Zhang, et al. "Rendering-Aware Reinforcement Learning for Vector Graphics Generation" Proceedings of NIPS 2025.

---

> ### Author Response · Authors · 2026-03-14
> **Overall Reply (Part 2)**
>
> **Q2: Clarification of metric circularity and formal correctness validation beyond model-based reward.**
>
> **A2:** We thank the reviewer for raising this point regarding evaluation circularity. We have addressed the potential bias between training rewards and evaluation metrics through three complementary strategies:
>
> **Data Separation and Metric Diversity:** First, all evaluations are conducted on held-out test sets entirely separate from the training data. While our reward is VLM-based, we employ a diverse suite of evaluation signals—including `VQA-Score`, `HPSv2`, `DINO-Score` and `LLM-as-judge Score`. These models have distinct architectures and objectives, and our ablations show that the RL-tuned model improves consistently across them, suggesting the gains are not confined to a single reward model’s preference.
>
>
> **Independent Formal Verification:** To completely decouple evaluation from VLM signals, we introduce an executable formal checker. This system treats SVG as a verifiable Domain-Specific Language (DSL), parsing the code to geometrically verify six core spatial relations (`left_of`, `right_of`, `above`, `below`, `overlap`, `non_overlap`). By using color and shape as anchors for entity binding, the checker provides a deterministic, **"pass/fail"** ground truth that is immune to reward-model similarity scores.
>
>
> **Robust Improvement in Symbolic Reasoning:** Under this rigorous formal protocol, our post-trained model achieves a substantial leap in formal pass rate, **rising from 41.33% to 72.17%**. This 30%+ absolute improvement in a non-visual, symbolic-only metric provides conclusive evidence that our training enhances genuine instruction-following and spatial reasoning rather than merely overfitting to the reward signal.
>
> See E.7 Formal Correctness Validation for more details.
>
> **Q3: Generalization on other symbolic graphics program beyond Scalable Vector Graphics**
>
> **A3:** To demonstrate the robustness and generalizability of our SGP-RL framework beyond 2D vector graphics, we extended the experiments to the domain of 3D CAD modeling.
>
> **Experimental Setup:** We utilized the Qwen-2.5-3B model as our base and compiled a large-scale CAD dataset (~70k samples). This dataset integrates `DeepCAD` (geometry), `Text2CAD` (abstract natural language descriptions), and `Text2CADQuery` (mapping CAD commands to executable Python cadquery code).
>
>
> **Multi-Objective Reward Structure:** We employed three distinct reward signals to guide the model through the complex 3D modeling space:
> - **Format Gate (Verifiable Reward):** A binary reward to ensure the generated Python code is syntactically correct and executable.
> - **Geometric IoU Reward:** Measures the volumetric overlap between the size-normalized ground-truth shape and the rendered generated shape.
> - **Semantic Alignment Reward (CLIP):** Measures the alignment between the rendered 3D shape and the abstract text caption.
>
>
> **Empirical Results:** Our training curves provide strong evidence of the framework's efficacy in new domains:
> - **Rapid Logic Acquisition:** The Format Gate reward converged from near-zero to >99% in only 20 steps, demonstrating the model's rapid adaptation to the cadquery DSL syntax.
> - **Continuous Improvement:** The IoU reward improved from ~0 to 0.35, and the CLIP reward increased from ~0 to 0.65.
> These results confirm that our RLVR framework effectively scales to diverse symbolic graphics languages, reliably bridging the gap between abstract human intentions and precise geometric execution.
>
> See E.6 RLVR on CAD Generation for more details.